# Removal of inhibition uncovers latent movement potential during preparation

Uday K Jagadisan[1,2]*, Neeraj J Gandhi[1,2,3,4]

[1]Department of Bioengineering, University of Pittsburgh, Pittsburgh, United States; [2]Center for the Neural Basis of Cognition, University of Pittsburgh, Pittsburgh, United States; [3]Department of Otolaryngology, University of Pittsburgh, Pittsburgh, United States; [4]Department of Neuroscience, University of Pittsburgh, Pittsburgh, United States

**Abstract** The motor system prepares for movements well in advance of their execution. In the gaze control system, the dynamics of preparatory neural activity have been well described by stochastic accumulation-to-threshold models. However, it is unclear whether this activity has features indicative of a hidden movement command. We explicitly tested whether preparatory neural activity in premotor neurons of the primate superior colliculus has 'motor potential'. We removed downstream inhibition on the saccadic system using the trigeminal blink reflex, triggering saccades at earlier-than-normal latencies. Accumulating low-frequency activity was predictive of eye movement dynamics tens of milliseconds in advance of the actual saccade, indicating the presence of a latent movement command. We also show that reaching a fixed threshold level is *not* a necessary condition for movement initiation. The results bring into question extant models of saccade generation and support the possibility of a concurrent representation for movement preparation and generation.
DOI: https://doi.org/10.7554/eLife.29648.001

*For correspondence:
kj.udayakiran@gmail.com

Competing interests: The authors declare that no competing interests exist.

## Introduction

The ability to interact with the world through movements is a hallmark of the animal kingdom. Movements are usually preceded by a period of planning, when the nervous system makes decisions about the optimal response to a stimulus and programs its execution. Such planning behavior is seen in a wide variety of species, including, insects (*Fotowat and Gabbiani, 2007*; *Card and Dickinson, 2008*), fish (*Preuss et al., 2006*), frogs (*Nakagawa and Nishida, 2012*), and mammals (*Hanes and Schall, 1996*; *Churchland et al., 2006a*). A fundamental question in sensorimotor neuroscience is how planning activity is appropriately parsed in order to prepare and execute movements.

In the primate gaze control system, premotor neurons that produce a volley of spikes to generate a movement are typically also active leading up to the movement. Build-up of low frequency activity prior to the saccade command, or its signatures, have been observed in a wide variety of brain regions involved in gaze control, including the frontal eye fields (*Hanes and Schall, 1996*; *Gold and Shadlen, 2000*), lateral intraparietal area (*Platt and Glimcher, 1999*), and superior colliculus (SC) (*Dorris et al., 1997*). Since variability in the onset and rate of accumulation of low-frequency activity is correlated with eventual saccade reaction times (*Hanes and Schall, 1996*; *Ratcliff and Rouder, 1998*; *Usher and McClelland, 2001*), it is thought that this activity primarily dictates *when* the movement is supposed to be initiated. However, it is unclear how (or if) downstream motor networks distinguish activity related to movement preparation from the command to execute one.

One possibility is that the activity of premotor neurons undergoes a transformation from representing preparation into movement-related commands at a discrete point in time (*Thompson et al., 1996*; *Juan et al., 2004*; *Schall et al., 2011*), when the activity reaches a movement initiation

**eLife digest** Most of us are familiar with the experience of revving up the engine in anticipation of a red traffic light turning green. The revving, which enables us to move forward as soon as the brakes are released, reflects our ability to plan actions in advance. The brain shows broadly analogous behavior when preparing to move parts of the body. A few hundred milliseconds before we move our eyes, for example, brain regions responsible for eye movements start to gradually ramp up their activity.

Returning to the car analogy, revving up the engine will not cause the vehicle to move until we also release the brakes. In the brain, inhibitory mechanisms analogous to a brake prevent the increasing neural activity from triggering movement. The brain is thought to apply the brakes until this preparatory activity reaches a threshold, at which point the activity becomes a command to move. This raises the question: would the preparatory activity be able to trigger movement before reaching threshold if the brakes were no longer being applied?

To find out, Jagadisan and Gandhi devised experiments in which they could essentially "release the brakes" at any time. Monkeys were trained to stare at a central spot on a screen and, only when this spot disappeared, to then move their eyes to a target that appeared elsewhere on the screen. To release the brakes, Jagadisan and Gandhi used a puff of air to make the monkeys blink reflexively. Reflex blinks turn off the inhibition, which in the case of eye movements originates in a structure called the brainstem. This in turn enabled the monkeys to move their eyes while the neural activity was still ramping up. Further analysis showed that preparatory activity in another region of the brain that sends signals to the brainstem – the superior colliculus – predicted the speed of the resulting eye movement.

Together these results show that the neural activity involved in planning movements also has the potential to generate movement when released from inhibition. Understanding how the brain starts to produce a movement will allow scientists to probe why this process sometimes goes awry, for example during impulsive movements in ADHD and schizophrenia. It should also help decode the patterns of activity that the brain uses to represent movements before those movements occur. This could lead to improvements in technologies that enable patients to use brain activity to control artificial limbs.

DOI: https://doi.org/10.7554/eLife.29648.002

criterion, thereby acquiring the <u>potential</u> to generate a movement. Indeed, this is the basis for stochastic accumulator models of saccade initiation, in which premotor activity must reach a threshold level in order to generate the movement (*Hanes and Schall, 1996*; *Ratcliff and Rouder, 1998*; *Zandbelt et al., 2014*). Recent work in the skeletomotor system has also suggested that neuronal population activity undergoes a state space transformation just prior to the movement, thus permitting movement preparation without execution before the transformation (*Churchland et al., 2012*; *Kaufman et al., 2014*; *Elsayed et al., 2016*). These related views dictate that movement planning and execution are implemented as serial processes in the motor system. Alternatively, it is possible that neurons in sensorimotor structures represent these signals concurrently, gearing up to execute a movement in proportion to the strength of the planning activity. In other words, preparatory build-up of neural activity can multiplex higher order signals while simultaneously relaying those signals to effectors; we call this latter property the 'motor potential' of preparatory activity. This idea is in fact the premise of the premotor theory of attention (*Rizzolatti et al., 1987*; *Hoffman and Subramaniam, 1995*), and represents a latent behavioral manifestation of movement preparation.

How might we test for the presence of a motor potential in low frequency preparatory activity? The following thought experiment helps illustrate one approach. Consider the activity of a premotor neuron accumulating over time. Under normal circumstances, inhibitory gating on the saccadic system is released at an internally specified time, possibly when activity crosses a purported threshold level or when the population dynamics reach the optimal subspace, thus resulting in movement generation (top row in *Figure 1a*). We can then infer that the high frequency burst during movement execution has motor potential if neural activity is correlated with dynamics of the ensuing movement, that is, saccade velocity is faster when burst activity is higher, and vice versa (match gray traces

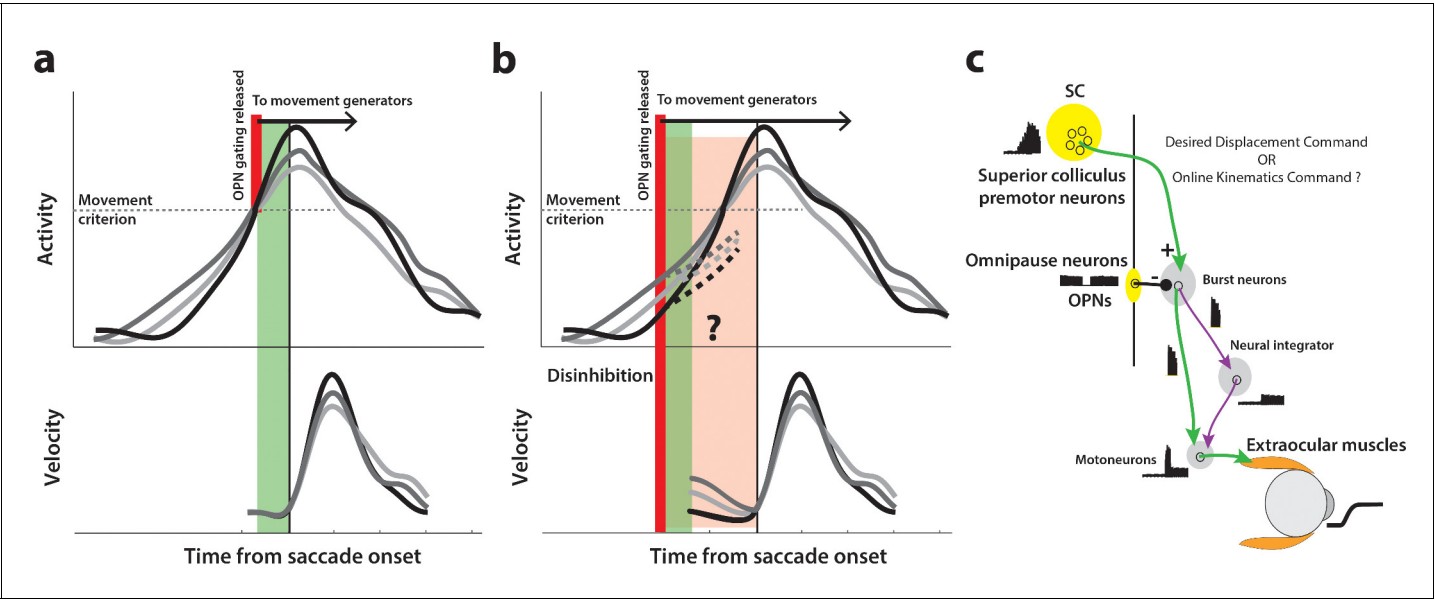

**Figure 1.** Conceptual schematic of saccade and pre-saccade motor potential revealed by disinhibition. (a) Under normal conditions, premotor activity accumulates at different rates on different trials (three example traces in top row) to a movement initiation criterion, opening downstream gating (thick red line) and triggering the saccade following an efferent delay (~20 ms, green window). Following saccade onset, variation in neural activity is correlated with variation in saccade velocity (match gray scale in top and bottom rows), indicating the neuron's motor potential. (b) Removal of inhibition through an experimental manipulation at an earlier time (thick red line), during saccade preparation and before the typical movement criterion is reached. The disinhibition may reveal the motor potential of ongoing activity in the form of correlated kinematics of the eye before onset of the actual saccade (light red region, velocity traces in bottom row), and also allows us to study any changes in the dynamics of activity leading up to the reduced latency saccades (dashed activity traces in top row). (c) Schematic of the premotor circuitry involved in saccade generation. A desired displacement command (or, as tested here, a kinematics-driving command) is sent from neurons in SC to the burst neurons in the brainstem reticular formation (also referred to as the burst generator). This pathway is gated by tonic inhibition from the OPNs under normal conditions. Approximately 20 ms before saccade onset, OPN activity pauses, disinhibiting the burst neurons and allowing the excitatory pathway (green arrows) to actuate an eye movement. This study tests whether disinhibiting the pathway downstream of SC by blink-induced suppression of OPNs results in an immediate eye movement.

DOI: https://doi.org/10.7554/eLife.29648.003

between top and bottom rows in *Figure 1a*). Now, if inhibition was somehow removed at a prior time through an experimental manipulation instead of allowing the system its natural time course (thick red line in *Figure 1b*), the occurrence of an early movement would indicate that ongoing low frequency preparatory activity <u>also</u> possesses motor potential. Importantly, this potential can be quantified by correlating neural activity with kinematics of the eye *before* the onset of the saccade proper (pre-saccade velocity traces in *Figure 1b*), and comparing it to the previously estimated potential after saccade onset. Furthermore, the dynamics of activity following the manipulation would indicate whether the activity must cross a decision boundary (i.e., threshold or optimal subspace) in order to produce the movement (dashed traces in *Figure 1b*). This hypothetical manipulation would therefore simultaneously shed light on both concurrent processing of preparatory signals and the criterion for movement initiation.

In this study, we used the trigeminal blink reflex to remove inhibition on the gaze control network during ongoing low-frequency activity. The omnipause neurons (OPNs) in the brainstem, which discharge at a tonic rate during fixation and are suppressed during saccades (*Figure 1c*, *Cohen and Henn, 1972*; *Keller, 1974*), also become quiescent during eye movements associated with blinks (*Schultz et al., 2010*). Previous work in our lab has shown that removal of this potent source of inhibition on the saccade burst generator with reflex blinks triggers saccades at lower-than-normal latencies (*Gandhi and Bonadonna, 2005*), an observation that has been used to study latent sensorimotor processing in SC (*Jagadisan and Gandhi, 2016*), the motor potential of a target selection signal during visual search (*Katnani and Gandhi, 2013*), and the dynamics of movement cancellation during saccade countermanding (*Walton and Gandhi, 2006*). Here, we first established that the saccade-related burst in SC has motor potential under normal conditions, by correlating the activity

during the burst to saccade kinematics on individual trials. Critically, when performing the same analysis in the perturbation condition, we found that the level of preparatory activity at the time of the blink was also strongly correlated with initial dynamics of the evoked movement, *prior* to the saccade proper, suggesting that ongoing sub-threshold activity in SC also possesses motor potential. Finally, we show that although these movements were preceded by an acceleration of ongoing activity following the perturbation, it is not necessary for preparatory activity in SC to reach a fixed threshold level before a saccade is produced – neural activity just prior to saccades triggered by reflex blinks was lower at both individual neuron and population levels.

## Results

In order to explicitly test whether saccade preparatory activity contains a latent movement command, we transiently disinhibited the motor system during the preparatory period in monkeys performing the delayed saccade task. Briefly, each subject fixated on a central fixation point while a saccade target appeared in the periphery. After a random delay interval, the fixation spot disappeared, which was the cue (go cue) for the animal to make a saccade to the target. We induced reflex blinks during this preparatory epoch - after the go cue and before saccade typical reaction times. A reflex blink is a suitable perturbation because it removes inhibition on the saccadic system by turning off the OPNs and triggers gaze shifts at lower-than-normal latencies (*Gandhi and Bonadonna, 2005*). We first describe important aspects of the blink technique here. When induced during fixation in the absence of any other target, a reflex blink is accompanied by a blink-related eye movement (BREM) – the eyes turn nasally and downward before returning to the original fixation position in a loop-like trajectory (e.g., *Rottach et al., 1998*). Gaze shifts triggered by the blink in the presence of a peripheral target thus have the BREM component in addition to the saccade directed towards the target; we refer to them as blink-triggered movements or blink-triggered saccades. *Figure 2a* shows example velocity profiles of the BREM (thin gray traces in the left column) and the velocity profiles and spatial trajectories of three blink-triggered movements (colored traces in left and right columns, respectively) from one session. We computed onset times of saccades embedded in blink-triggered movements using a previously used model-free approach that is agnostic to any possible interaction between the BREM and saccade components (*Katnani and Gandhi, 2013*). Saccade onset was determined as the time at which the movement velocity on a given trial deviated (colored circles) from the expected BREM profile distribution (thick black traces in *Figure 2a*; only the mean BREM trace is shown in the right panel for clarity) for that session. As seen from the example blink-triggered movements, there was considerable variation in the time at which the eye movement deviated from the BREM profile towards the saccade goal, marking saccade onset. *Figure 2b* shows the distribution of saccade onset times relative to blink time obtained using this approach (for more details, see Methods – movement detection). It is worth noting here that the bimodality apparent in the distribution of saccade onset times in *Figure 2b* likely reflects the divide between trials in which the process behind saccade initiation was already underway by the time of the blink (delays <20 ms, to the left of the vertical black line), and trials in which the saccade was triggered *due* to disinhibition by the blink (delays >20 ms, to the right of the vertical black line). Since most analyses in this study were focused on processes occurring before saccade onset, we split the data from perturbation trials into two sets, along the aforementioned divide. Blink-triggered movements with saccade onset greater than 20 ms after blink onset, since they were likely triggered *ab initio* by blink-related disinhibition, are ideal for exploring whether pre-saccade preparatory activity has motor potential and to investigate how the dynamics of ongoing activity are affected by the perturbation. Therefore, we used this subset of trials for the motor potential and accumulation rate analyses. On the other hand, blink-triggered movements with saccade onset less than 20 ms after blink onset are likely cases in which the saccade initiation process was well under way or imminent, and are suitable for studying whether initiation can happen even before activity reaches a fixed criterion level. Thus, all blink-triggered movements were used for the threshold analysis (for more details, see Methods – inclusion criteria).

### The blink perturbation triggers low latency, but accurate, saccades

First, as one way to assay the behavioral manifestation of motor preparation, we verified that reflex blinks during the preparatory period produced low-latency saccades. *Figure 3a* shows saccade

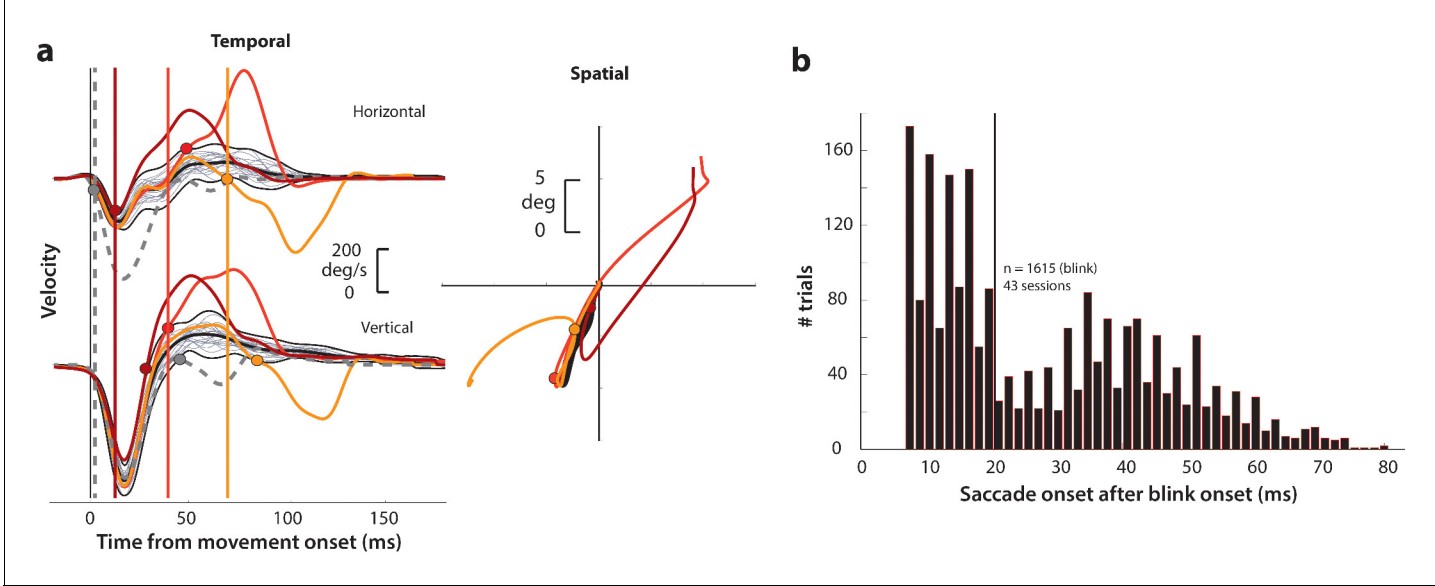

**Figure 2.** Determining goal-directed saccade onset in blink-triggered eye movements. (a) Left column: horizontal (top row) and vertical (bottom row) eye velocity profiles during blink-related eye movements obtained during fixation (BREMs, thin gray traces) and three example blink-triggered saccadic eye movements (colored traces) from one session. The thick black trace in the middle of the BREM profiles is their mean and the two black traces above and below it are ±2.5 s.d. bounds. Saccade onset was determined to be the point where an individual velocity profile (colored traces) crossed the BREM bounds and stayed outside for 15 consecutive time points. This point was determined independently for the horizontal and vertical channels (colored circles corresponding to each trace in the two rows), and the earlier time point was taken as the onset of the overall movement (vertical colored lines). The dashed gray velocity profile that deviates from the BREM very close to movement onset (<5 ms) is shown to highlight a case where the saccade starts before being perturbed by the blink, and was not considered as a blink-triggered movement in this study. Right column: Spatial trajectories of the eye for the three example blink-triggered movements from the left column, with the corresponding saccade onset time points indicated by the circles. The trajectory of the BREM template is shown in black (for the sake of clarity, only the mean is shown). Note that the movements in this session were made to one of two possible targets on any given trial. (b) Distribution of goal-directed saccade onset times relative to overall movement onset for blink-triggered movements (n = 1615) across all sessions (n = 43). For the motor potential and accumulation rate analyses (*Figures 4–5* and *7*), we only considered movements where the blink-triggered saccade started at least 20 ms after overall movement onset (to the right of the vertical line). For the threshold analysis (*Figure 6*), we considered all blink-triggered movements.
DOI: https://doi.org/10.7554/eLife.29648.004

reaction time (from GO cue) as a function of the time of blink across all perturbation trials (red circles). To visually compare reaction times on blink trials with those in control trials, it was necessary to include the distribution of control reaction times in this figure. To do this, we created a surrogate dataset by randomly assigning blink times to control trials, and plotted them on the same axes as blink trials in *Figure 3a* (blue circles). Reaction times in perturbation trials were correlated with time of blink, and were significantly lower than control reaction times (mean control reaction time = 278 ms, mean blink-triggered reaction time = 227 ms, p=2.2×10$^{-197}$, one-tailed t-test), consistent with previous observations (*Gandhi and Bonadonna, 2005*).

We then verified whether saccades triggered by the blink were as accurate as normal saccades, in order to eliminate any potential confounds due to differences in accuracy. We calculated saccade accuracy as the Euclidean endpoint error normalized with respect to target location (inset in *Figure 3b*). The distributions of relative errors for all control and blink trials are shown in *Figure 3b*. Blink-triggered saccade accuracy was not significantly different from control saccades (mean control accuracy = 0.136, mean blink-triggered accuracy = 0.133, p=0.3, two-tailed t-test), as reported previously (*Goossens and Van Opstal, 2000b*; *Gandhi and Bonadonna, 2005*). We also tested for and found no relationship between blink time and saccade accuracy (Spearman's correlation = 0.04, p=0.22). Note that the eyes are closed due to the blink, so visual feedback does not contribute to endpoint accuracy. Nonetheless, we have demonstrated previously that blanking the visual target during the blink-triggered movement does not compromise the accuracy on perturbation trials (*Gandhi and Bonadonna, 2005*). The blink perturbation thus provides an assay to study the question of motor potential without introducing confounding factors related to saccade metrics.

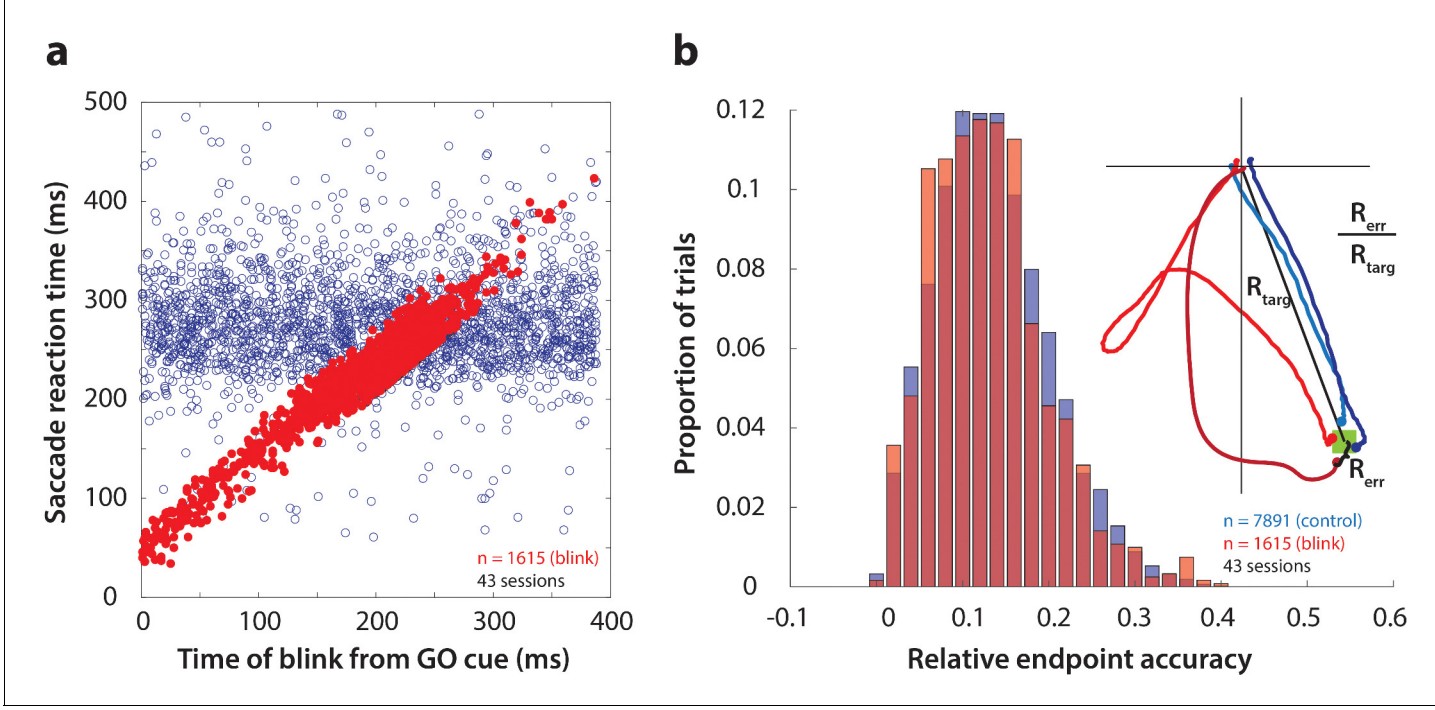

**Figure 3.** Time course and accuracy of blink-triggered saccades. (**a**) Saccade reaction time as a function of blink onset time across all trials (n = 7891 control trials, 1615 blink trials) and sessions (n = 43). Red filled circles are individual blink trials, and blue circles are from control trials with randomly assigned blink times for comparison. For visualization alone, only a random selection of control trials equal in number to blink trials are plotted. (**b**) Endpoint accuracy for control and blink-triggered movements (blue and red histograms, respectively) across all trials from all sessions. To enable comparison across movement amplitudes, the endpoint error was normalized as the actual Euclidean endpoint error from the target divided by the eccentricity of the target, as depicted in the spatial plot in the inset.
DOI: https://doi.org/10.7554/eLife.29648.005

## SC activity is correlated with saccade kinematics for normal saccades

Next, as a crucial prerequisite for our motor potential analysis, we examined how the motor burst in SC is correlated with saccade kinematics. Specifically, we computed the correlation between the trial-by-trial firing rates of a neuron and the corresponding velocities. This approach to estimating motor potential is illustrated in *Figure 4a*. Since our eventual goal was to study pre-saccade motor potential in blink-triggered saccades, we used the component of velocity in the direction of the saccade goal as our kinematic variable on a given trial (inset in *Figure 4a*, see Methods – kinematic variables, for details). The choice to use projected kinematics is to maintain uniformity with analyses on blink-triggered movements (see next section), but performing the analysis on the raw, unprojected kinematics for control saccades yielded very similar results (*Figure 4—figure supplement 1*). Further, to avoid assumptions about the efferent delay between SC activity and ocular kinematics, we computed the activity-velocity correlation at various potential delays between the two signals (the blue bars in *Figure 4a* show an example delay of 12 ms – compare the similarly shaded bars in the velocity and activity panels). *Figure 4b* shows, for an example neuron, the trial-by-trial scatter of velocities at three time points (15 ms before, at, and 15 ms after saccade onset – light, medium, and dark blue circles, respectively) plotted against neural activity preceding those respective velocities by 12 ms. Note that a strong across-trial correlation between activity and kinematics is seen only at the time point that is after saccade onset in this example.

It is not surprising to see a lack of correlation with SC activity for pre-saccade velocities, since they are largely zero or constant, by definition, for normal saccades, because the inhibitory gating by OPNs has not been removed yet. In contrast, the strong correlation between kinematics and activity following movement onset indicates the presence of a motor potential in the saccade-related burst. We systematically explored the time course of this motor potential by computing the correlation at different time points before and during the saccade, for a range of delays between activity

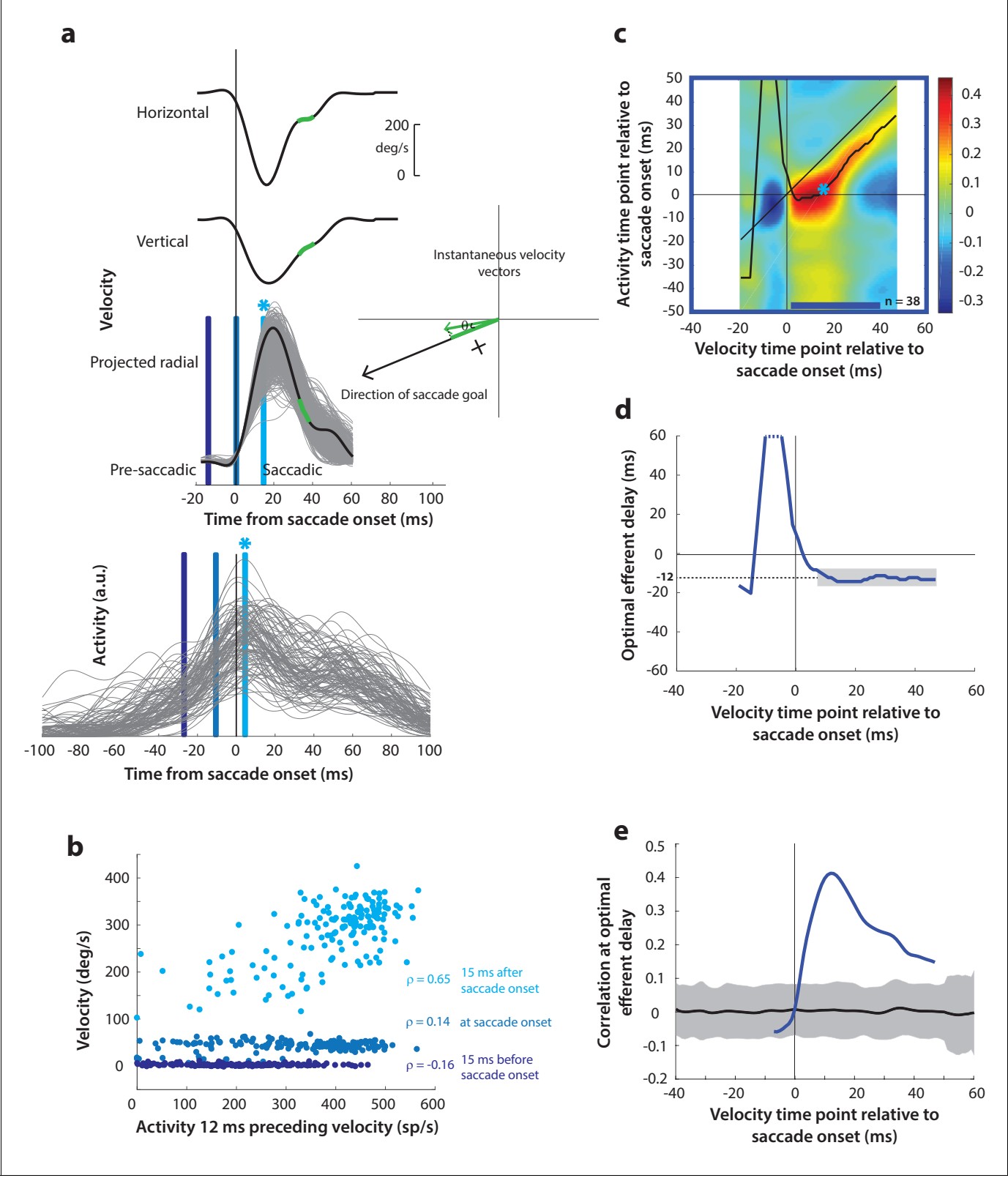

**Figure 4.** Motor potential during control saccades. (a) Estimating motor potential as correlation between neural activity and saccade kinematics. Horizontal and vertical velocity traces (top two rows) on control trials are converted to radial velocity (third row) in the direction of the saccade goal. The projection of the green vector in the inset and the corresponding green parts of the velocity traces illustrate this computation. One trial (thick black trace) is highlighted for clarity. For control trials, the projected and executed vectors are very similar. The bottom row shows neural activity traces on

*Figure 4 continued on next page*

*Figure 4 continued*

different trials for the neuron recorded in this example session. (b) Motor potential is estimated as the correlation between neural activity and projected saccade kinematics. The scatter plot of the projected radial velocity 15 ms after saccade onset, at saccade onset, and 15 ms before saccade onset (light, medium, and dark blue points, respectively, and corresponding vertical lines in panel a) against neural activity 12 ms preceding the velocity time points shows that the neuron has motor potential once the saccade has started (Pearson's correlation coefficient = 0.65). Each point corresponds to one trial. (c) Point-by-point correlation between velocity and activity, averaged across neurons. Heat map colors represent correlation values. As an example, the light blue asterisk refers to the correlation between the velocity and activity corresponding to the time points with the asterisk in panel a. The black curve traces the contour of the highest correlation time points in the activity for each point during the movement. The blue bar at the bottom of the heatmap indicates timepoints at which the average correlation was significant (based on ±95% CI from panel e). (d) Optimal efferent delay computed as the distance of the black trace in panel c from the unity line. Negative values for the delay are causal, i.e., correlation was high for activity points leading the velocity points. The shaded gray bar shows that the optimal delay was consistent during the movement (mean for shaded region = −12 ms) (e) Population average correlation as a function of time at the −12 ms estimated efferent delay. The black trace is the mean and the gray region is the ±95% confidence interval for the bootstrapped (trial-shuffled) correlation distribution.
DOI: https://doi.org/10.7554/eLife.29648.006

The following figure supplement is available for figure 4:

**Figure supplement 1.** Motor potential during control saccades, computed with raw velocities.
DOI: https://doi.org/10.7554/eLife.29648.007

and kinematics. We did this for each neuron, and the population average correlation coefficients at each time point and delay are shown in the heat map in *Figure 4c*. To aid interpretation of this figure, the light blue asterisk in the heat map refers to the correlation computed at the time points with the corresponding asterisk in *Figure 4a*. Motor potential of SC activity emerged only after the onset of the saccade, and lasted throughout the movement (streak of correlation below the unity line in *Figure 4c*).

For each time point in the velocity signal, we also computed the activity time points at which correlation was highest, shown as the running mean (black trace) in the heat map. This provides a measure of the efferent delay at which neural activity is most effective in driving movement kinematics. Note that the black trace is roughly parallel to the unity line, suggesting that the efferent delay was consistent for the duration of the movement. To characterize this property better, we plotted optimal efferent delay, calculated as the difference between the black trace and the unity line, as a function time with respect to movement onset (*Figure 4d*). The mean delay during the movement was −12.2 ms (±s.d. = 1.7 ms), meaning that instantaneous saccade kinematics were best predicted by SC activity roughly 12 ms before. This value is centered within the range of previous functional estimates of the conduction delay between SC and extra-ocular muscles (8–17 ms, *Miyashita and Hikosaka, 1996*). The values for the delay before saccade onset are highly variable, likely due to noise in the pre-saccade velocities, and therefore should be ignored (also note that these delays are positive and therefore non-causal). Finally, *Figure 4e* shows the correlation values at the −12 ms delay as a function of time. The gray region is the ±95% confidence intervals of the population average bootstrapped (trial-shuffled) distribution of correlations (see Methods – surrogate data and statistics). Thus, for normal trials, motor potential, in the form of correlation between neural activity and eye velocity, only manifests after the onset of the saccade proper (starting 3 ms after saccade onset, p<0.05, bootstrap test).

## SC preparatory activity preceding the saccade-related burst possesses motor potential

Having found a correlation between SC activity and ocular kinematics during saccades, we wanted to know whether the time course of such motor potential expands when the saccadic system is disinhibited at earlier time points. Specifically, we wanted to know whether ongoing preparatory activity contained any motor potential before its maturation into a motor command. We hypothesized that, if the low-frequency preparatory activity that precedes the high-frequency burst had motor potential, removing inhibitory gating on the system would result in a slow eye movement proportional to the level of activity before accelerating into a saccade. The rationale was that, because the downstream OPNs become quiescent during the blink, activity in SC neurons that possessed motor potential would be allowed to drive the burst generators, and subsequently, the eye muscles (see circuit diagram in *Figure 1c*). To test this, we computed the correlation between SC activity and eye velocity before and during blink-triggered saccades in a manner like that for control saccades

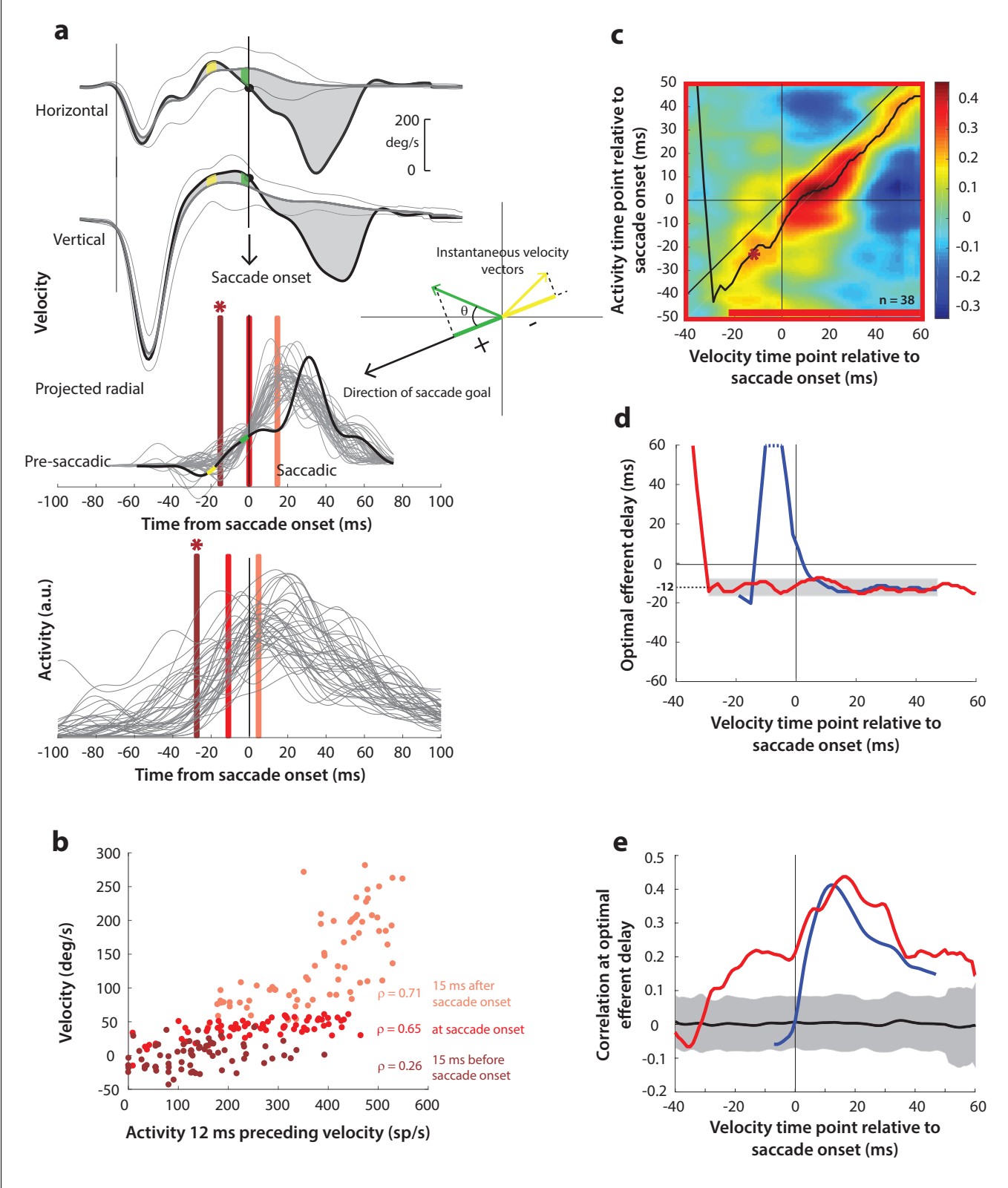

**Figure 5.** Motor potential on blink trials. (a) Computation of the kinematic variable for blink-triggered movements. Horizontal and vertical velocities (thick black traces in top two rows) are converted to residual velocities (gray fill) after subtracting the corresponding mean BREM template (middle thin black trace in top two rows), in order to discount the effects of intrinsic variability in the BREM itself. The radial residual velocity (third row) in the direction of the saccade goal is then computed, similar to *Figure 4a*. The green and yellow time points in the velocity traces (shown as thicker than an

*Figure 5 continued on next page*

*Figure 5 continued*

instant for clarity), represented as corresponding velocity vectors in the inset, illustrate this process. For example, the green velocity residual, immediately before saccade onset, deviates negatively from the BREM in the horizontal component, and positively in the vertical component, resulting in an instantaneous kinematic vector pointing leftwards and upwards. The component of this vector in the direction of the saccade goal is then taken as the kinematic variable for this time point (also compare *Figure 5—figure supplement 1*). (b) Scatter plot of the neural activity versus velocity at the three indicated time points (red points of various saturation, at corresponding red lines in panel a) for blink-triggered movements. As in *Figure 4b*, these are plotted at the −12 ms shift between activity and velocity Note the strong correlation for pre-saccade time points compared to *Figure 4b*. (c) Point-by-point correlation of projected residual velocity with activity, averaged across neurons, for blink-triggered movements. The velocity time points are with respect to time of saccade onset extracted from the blink-triggered movement. The dark red asterisk points to the correlation between the velocity and activity corresponding to the time points with the asterisk in panel a. The black curve traces the contour of the highest correlation time points in the activity for each point during the movement. The red bar at the bottom of the heatmap indicates timepoints at which the average correlation was significant (based on ±95% CI from panel e). (d) Optimal efferent delay computed as the distance of the black trace in panel c from the unity line. The red trace is for blink-triggered movements, and the blue trace is from *Figure 4d* for control saccades, overlaid for comparison. Negative values for the delay are causal, i.e., correlation was high for activity points leading the velocity points. The gray bar highlights the fact that the optimal delay was consistent during both control and blink-triggered saccades, and both before and after saccade onset for the latter (mean for shaded region = −12 ms). (e) Population average correlation for blink-triggered movements (red trace) as a function of time at the −12 ms estimated efferent delay. The blue trace is from *Figure 4e* for control saccades, overlaid for comparison. The black trace is the mean and the gray region is the ±95% confidence interval for the bootstrapped (trial-shuffled) correlation distribution.

DOI: https://doi.org/10.7554/eLife.29648.008

The following figure supplements are available for figure 5:

**Figure supplement 1.** Motor potential on blink trials, computed with raw velocities.
DOI: https://doi.org/10.7554/eLife.29648.009
**Figure supplement 2.** 'Main sequence' velocity-amplitude relationship.
DOI: https://doi.org/10.7554/eLife.29648.010

(*Figure 5a*). It is important to note that we subtracted the BREM component from the blink-triggered saccade velocities before projecting these residuals in the direction of the saccade goal (inset in *Figure 5a*). This was done to prevent independent variations in BREM kinematics (unrelated to SC activity) from masking any underlying motor potential-related correlation, which we found might be the case when we performed this analysis on the raw velocities for blink-triggered saccades (*Figure 5—figure supplement 1a–b*). *Figure 5b* shows an example scatter plot of the trial-by-trial activities versus projected residual velocities for three time points with respect to saccade onset. In contrast with *Figure 4b*, activity was correlated with velocity at all three time points shown, including the ones at and 15 ms prior to saccade onset.

We then computed the population average correlation at different time points before and after the onset of the high velocity saccade, at different efferent delays. Neural activity was highly correlated with movement kinematics after the onset of the high velocity saccade component (after time 0 on the x-axis in the heat map in *Figure 5c*), in agreement with data from control saccades. Importantly, activity was also correlated with eye kinematics <u>before</u> saccade onset (time points before 0 on the x-axis in *Figure 5c*), suggesting that upstream SC activity leaked through to the eye muscles as soon as the OPNs were turned off by the blink, causing activity-related deviations in the kinematics around the BREM. The black trace in *Figure 5c* also shows that the estimated efferent delay was similar to that observed in control trials and consistent before and after saccade onset. This is better observed in *Figure 5d* (red trace) – the mean delay after saccade onset was −11.9 ms (±s.d. = 2.3 ms), not significantly different from the mean delay before saccade onset (mean ±s.d. = −11.8 ±1.9 ms, up to 30 ms before saccade onset; t-test, p=0.93). The time course of efferent delay estimates for control trials from *Figure 4d* is also overlaid for comparison in *Figure 5d* (blue trace). *Figure 5e* shows the correlation as a function of time during the blink-triggered movement at these mean pre- and post- saccade optimal delays (red trace), with the time course for control trials from *Figure 4e* overlaid for comparison (blue trace). The average correlation across the population was significant (red trace above gray distribution, p<0.05, bootstrap test) starting 30 ms before onset of the saccadic component and lasting until the end of the movement, reiterating the key result of the study.

Recall that, for control saccades, it made no difference whether the motor potential was computed based on the raw velocity traces or the component of velocity in the direction of the saccade (compare *Figure 4* and *Figure 4—figure supplement 1*), an expected result since instantaneous eye velocity is predominantly in the direction of the saccade goal during control saccades. In contrast,

for blink-triggered saccades, the motor potential was revealed only when considering kinematics in the direction of the saccade goal (*Figure 5*), and not with the unprojected kinematics (*Figure 5—figure supplement 1*). This observation is important for two reasons. First, it ensures that the pre-saccade motor potential seen during blink-triggered saccades is not a result of misestimating saccade onset within blink-triggered movements. If the observed pre-saccade correlation resulted solely from estimating saccade onset to be later than the ground truth, that is, it is actually a peri-saccade correlation in disguise, then it should persist even with the raw, unprojected blink-triggered velocity residuals, and at the same efferent delay as for the peri-saccade correlation. This is because we have already seen that motor potentials exist once the saccade has started. Second, it adds support to the notion of motor potential itself: if spikes from the preparatory activity of these neurons leaked through to the muscles, you would expect them to only drive kinematics in the neurons' preferred direction (as opposed to a non-selective impact on all movements).

Additionally, we also verified whether the observed speeds of the pre-saccadic and saccadic components of blink-triggered movements were consistent with the neurons' position along the rostro-caudal extent of the SC. The number of terminating boutons from SC neurons to the horizontal burst generator has been shown to increase monotonically along the rostro-caudal axis of SC (*Moschovakis et al., 1998*). This increasing projection strength is reflected in the 'main sequence' of increasing saccade peak velocities with saccade amplitude (and therefore its rostro-caudal position) (*Gandhi and Katnani, 2011*). We reasoned that if the projection strength of a given neuron is fixed, then any kinematics, even pre-saccadic ones, caused by spikes emitted by that neuron must scale with the projection strength along the rostro-caudal axis. *Figure 5—figure supplement 2a* shows that the peak velocity of the saccade component in blink-triggered movements increased with amplitude of the eventual movement, re-confirming the well-known main sequence relationship for normal saccades. Consistent with this relationship and its implications for rostro-caudal projection strengths, the average pre-saccade velocities (in a 20 ms window before detected saccade onset) also showed a significant positive relationship with movement amplitude (*Figure 5—figure supplement 2b*), although this relationship was weaker due to the relatively bigger spread of pre-saccade velocities for a given amplitude. We also verified whether the previously computed across-trial correlations between activity and velocity varied as a function of rostro-caudal position, and did not find a significant relationship (not shown). This suggests that a strongly projecting neuron is as likely to have a correlation with kinematics as a weakly projecting one – but the actual kinematics are what are governed by the projection strengths.

Note that the time course of motor potential, when it exists, and estimated efferent delay, are remarkably similar across all conditions and analyses (*Figure 5—figure supplement 1d–e*), including the latent pre-saccade potential seen with the projected kinematics. Moreover, both subjects exhibited qualitatively similar properties in terms of pre-saccade motor potential and efferent delays, although the strength of the activity-velocity correlation was weaker in one of them (not shown). Put together, these observations strongly suggest pre-saccade preparatory activity indeed possesses motor potential which is revealed when the appropriate correlations between neural activity and kinematic variables are computed.

## Blink-triggered saccades are evoked at lower thresholds compared to normal saccades

Our results so far show that like the saccade-related burst, preparatory activity in superior colliculus also has movement-generating potential, which is normally hidden due to downstream inhibitory gating. Next, we used the fact that blink-triggered saccades are evoked at lower latencies to study the related question of what factors determine movement initiation under normal circumstances. Specifically, we sought to test an influential model of saccade initiation – the threshold hypothesis (*Hanes and Schall, 1996*). We asked whether it is necessary for activity in SC intermediate layer neurons to reach a fixed activity criterion in order to generate a movement. Previous studies have estimated the threshold for individual neurons in SC and FEF by assuming a specific time at which the threshold could be reached before saccade onset or by computing the time, backwards from saccade onset, at which premotor activity starts becoming correlated with reaction time (*Hanes and Schall, 1996*; *Paré and Hanes, 2003*). Given the heterogeneity of the activity profiles of premotor neurons, we think this approach is too restrictive to obtain an unbiased estimate of the threshold, if any. Instead, we took a non-parametric approach and scanned through possible times at which a

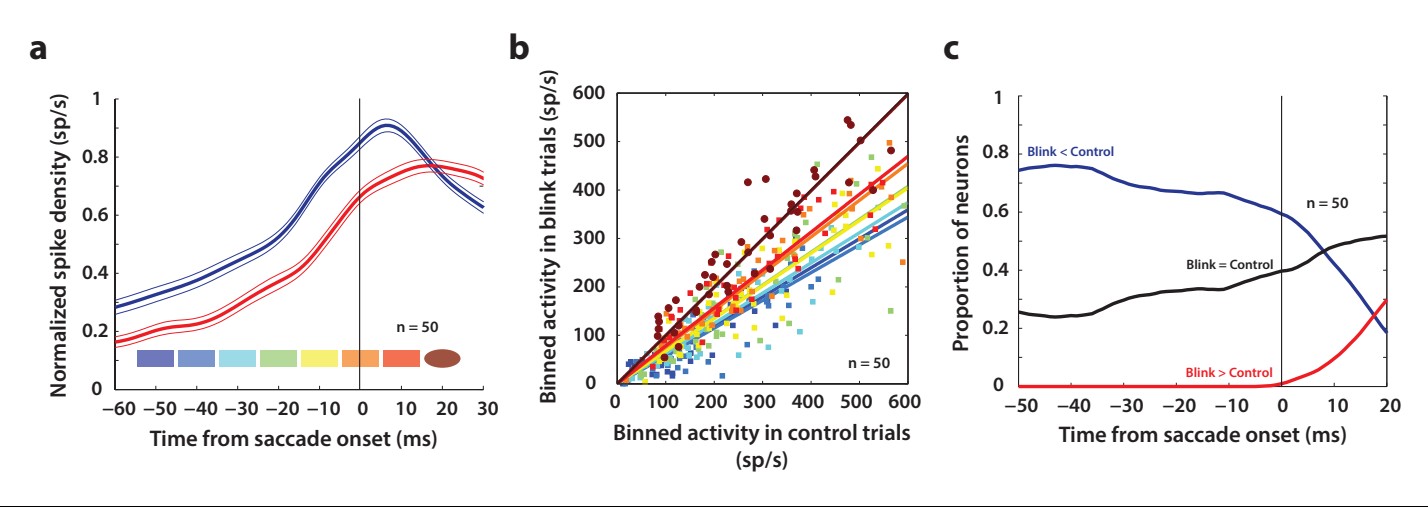

**Figure 6.** Analysis of putative threshold. (a) Average normalized population activity (thick traces) aligned on saccade onset for control (blue trace) and blink (red trace) trials. The thin lines represent s.e.m. The colored swatches at the bottom show the time windows used for the analysis presented in **b**; their shapes represent the presence (rectangle) or absence (ellipse) of a significant difference between control and blink rates in that time window. (b) Scatter plot of the activities of individual neurons (n = 50) in the time windows illustrated in (a) for control versus blink trials (each set of colored points corresponds to one time window). Square points indicate that the activity in control trials was higher in that window compared to blink trials, and circles indicate that there was no significant difference between the two conditions. Colored lines indicate linear fits to the scatter at the corresponding time window. The diagonal (thin black line) is the unity line; it overlaps with the dark brown line. (c) Proportion of neurons exhibiting the corresponding labelled differences between the two conditions as a function of time.

DOI: https://doi.org/10.7554/eLife.29648.011

putative threshold might be reached prior to saccade onset (*Jantz et al., 2013*). *Figure 6a* shows a snippet of the average population activity, normalized by the peak trial-averaged activity during control trials for each neuron, and aligned on saccade onset for control (blue traces) and blink (red traces) trials. For each neuron in this population (n = 50), we computed the average activity in 10 ms bins slid in 10 ms increments from 50 ms before to 20 ms after saccade onset (colored windows at the bottom of *Figure 6a*). If activity on control trials reaches the purported threshold at any one of these times before saccade onset, a comparison with activity in blink trials at that time should reveal the existence, or lack thereof, of a fixed threshold. *Figure 6b* shows the activity in these bins for control trials plotted against blink trials, colored according to the bins in *Figure 6a*. Note that the majority of points for early time bins lie below the unity line. Activity on blink trials was significantly lower compared to control activity from 50 ms before to 10 ms after saccade onset (square points, Wilcoxon signed-rank test, comparisons in each of these windows were significant at $p=10^{-6}$, at least). The systematic trend in the linear fits (solid lines) to these points suggests that the activity on blink trials gradually approaches that on control trials; however, the earliest time at which control activity was not different from activity in blink trials was 20 ms after saccade onset (circles, Wilcoxon signed-rank test, p=0.8) – too late to be considered activity pertaining to a movement initiation threshold. Thus, activity at the population level need not reach a fixed threshold level in order to produce a movement.

Nevertheless, we wanted to know if there exist individual neurons in the population that might obey the threshold hypothesis. For each neuron, we calculated whether activity on blink trials was higher, lower, or not significantly different from activity in control trials, at each time point from −50 before to 20 ms after saccade onset (Wilcoxon rank-sum test, comparisons in each of these windows were significant at p=0.001, at least). The three traces in *Figure 6c* represent the proportion of neurons that showed each of those three characteristics as a function of time. As late as 10 ms before saccade onset, more than 60% of the neurons had lower activity on blink trials compared to control trials (blue trace), inconsistent with the idea of a fixed threshold. Roughly 30% of the neurons did not exhibit significant differences in activity on blink and control trials at that time point (black trace); however, this observation is insufficient to conclude that the activities in the two conditions were identical, or that it must reach a threshold. Of course, it is possible that some of these neurons

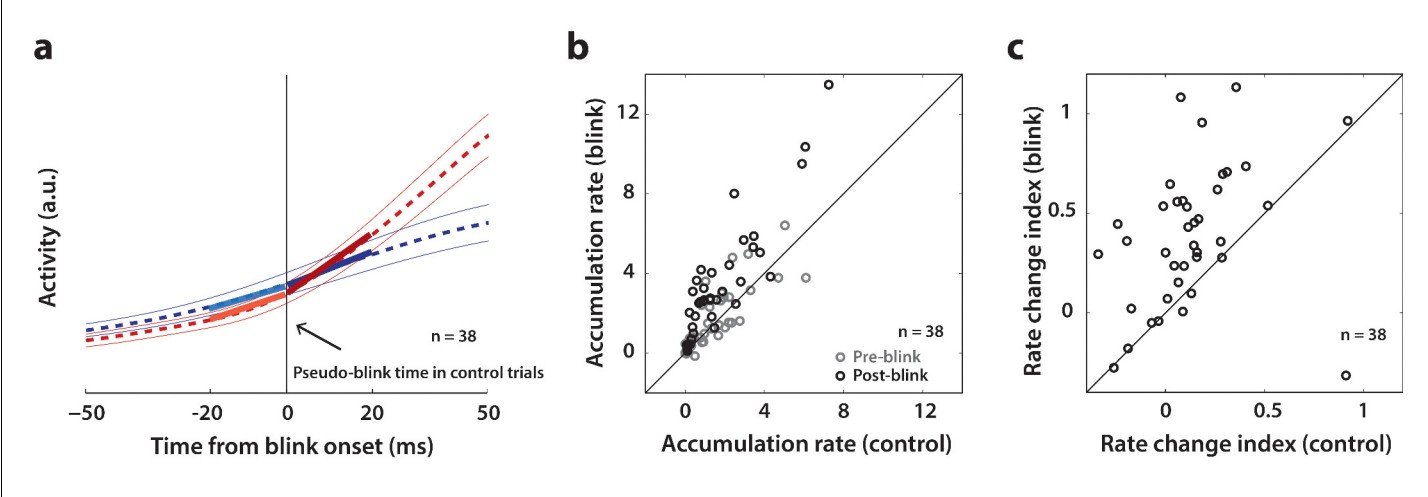

**Figure 7.** Analysis of accumulation rate change following perturbation. (**a**) Schematic illustrating the computation of accumulation rates before and after the blink. The snippet shows the average population activity (dashed lines) ± s.e.m (thin lines) centered on blink onset for blink trials (red trace) and pseudo-blink onset from the surrogate dataset for control trials (blue trace). The thick lines represent linear fits to the activity 20 ms before (lighter colors) and after (dark colors) the blink time. (**b**) Scatter plot of pre-blink (gray circles) and post-blink (black circles) accumulation rates (slopes of fits shown in panel a) of individual neurons for control versus blink trials. The unity line is on the diagonal. (**c**) Scatter plot of the pre-to-post rate modulation index $\frac{rate_{post}-rate_{pre}}{rate_{post}+rate_{pre}}$ for control versus blink trials.

DOI: https://doi.org/10.7554/eLife.29648.012

belong to a class for which fixed thresholds have been observed in previous studies. Together, these results suggest that it is <u>not necessary</u> for premotor activity in SC intermediate layers to reach a fixed threshold at the individual neuron or population level in order to produce a movement.

## Rate of accumulation of SC activity accelerates following disinhibition by the blink

Since SC neurons do not necessarily cross a fixed threshold to produce movements, as we saw above, it is possible that blink-triggered saccades are initiated directly off the ongoing level of preparatory activity. Alternatively, low frequency SC activity may be altered by the blink, even if the saccade is triggered at a lower level compared to control trials. Therefore, we studied whether the dynamics of SC activity are modulated by the blink prior to saccade initiation. Since we wanted to test for a change in dynamics before the actual saccade started, we restricted our analysis to the subset of trials in which saccade onset occurred at least 20 ms after blink onset. This restriction reduced our population to 38 neurons. For each neuron, we estimated the rate of accumulation of activity in 20 ms windows before and after blink onset with piecewise linear fits (*Figure 7a*, dashed red trace and solid lines). It is important to note that while the evolution of premotor activity is commonly modelled as a linear process, the actual dynamics of accumulation may be non-linear, causing spurious changes in linear estimates of accumulation rate over time. To account for this, we created a surrogate dataset of control trials for each neuron, with blink times randomly assigned from the actual distribution of blink times for that session. We then performed linear accumulation fits for the control dataset as well (dashed blue trace and solid lines in *Figure 7a*). Changes in accumulation rate on control trials following the pseudo-blink should reflect the natural evolution of activity at typical blink times and provide a baseline for comparing any changes observed in blink trials. *Figure 7b* shows a scatter plot of pre- and post- blink accumulation rates on control and blink trials. Pre-blink rates were not different between the two conditions (light circles, Wilcoxon signed-rank test, p=0.4), but post-blink rates were significantly higher on blink trials (dark circles, Wilcoxon signed-rank test, p=2.5×10$^{-6}$). Next, we tested for a change in accumulation rates following the blink by calculating a rate change index, defined as the difference of post- versus pre- blink rates divided by their sum, for each condition (*Figure 7c*). This index was positive for most neurons, even for control trials, highlighting the natural non-linear dynamics mentioned above. The change in accumulation rate was

significantly higher following the actual blink on blink trials (Wilcoxon signed-rank test, p=4.4×10$^{-6}$) compared to after the pseudo-blink on control trials. Thus, removal of inhibition seems to cause an acceleration in the dynamics of ongoing activity in the lead up to a saccade.

## Discussion

In this study, we sought to uncover the dynamics of movement preparation and, specifically, test whether the low-frequency preparatory activity of SC neurons encodes movement-related signals. We first established a baseline for this question by showing that, under normal conditions, saccade-related activity in the intermediate layers of SC has 'motor potential', defined as correlated variability between firing rate and saccade kinematics. Then, by disinhibiting the saccadic system much earlier than its natural time course with a reflex blink, we showed that low-frequency preparatory activity in these neurons also has a latent motor potential, indicating the presence of a hidden movement command. These results suggest that the output of higher order gaze control regions like the SC possesses motor potential during both low- and high-frequency activity, but a correlation with movement kinematics may only present itself when intermediary gating (in this case presented downstream of SC by the OPNs) between the two observables is turned off. We also found that SC activity does not necessarily have to reach a threshold at the single neuron or population level in order to initiate the saccade, contrary to the postulates of an influential model of saccade initiation – the threshold hypothesis (*Hanes and Schall, 1996*).

### Relationship between movement preparation and execution

Studies on the neural correlates of movement generation have largely focused on the divide between preparatory and executory activity, mainly due to the substantial natural latencies between the cue to perform a movement and its actual execution. Since variability in the properties of post-cue neural activity is correlated with eventual movement reaction times (*Hanes and Schall, 1996*; *Churchland et al., 2006b*), it is thought that this activity is purely preparatory in nature, influencing only *when* the movement is supposed to be initiated (*Kaufman et al., 2016*), and is devoid of the potential to generate a movement, until just before its execution. Indeed, there is some evidence that movement preparation and execution have distinct neural signatures with a well-defined boundary separating them. Studies of movement preparation in the skeletomotor system have shown that cortical activity reaches an optimal subspace before undergoing dynamics that produce a limb movement (*Afshar et al., 2011*; *Churchland et al., 2012*). A related idea suggests that activity pertaining to movement preparation evolves in a region of population space that is orthogonal to the optimal subspace, and this dissociation confers neurons the ability to prepare the movement by incorporating perceptual and cognitive information without risking a premature movement (*Kaufman et al., 2014*; *Elsayed et al., 2016*). However, in the absence of perturbations to the natural time course of movement generation, or explanatory models linking them to downstream processing, it is unclear whether such neural correlates are the sole causal links to the process of initiating movements.

How do we reconcile such previous observations with the finding in this study of a latent motor potential in preparatory activity? One possibility is that the oculomotor system operates differently from the skeletomotor system. More generally, it is possible that it is necessary for population activity to be in an optimal, 'movement-generating' subspace in order to release inhibition and/or effectively engage downstream pathways leading up to activation of the muscles, but once the motor system has been disinhibited by another means (e.g., the blink perturbation in this study), preparatory activity is read out as if it were a movement command. Based on the results in our study, we hypothesize that the activity is likely in the movement preparation subspace when the slow eye movement is produced after the blink perturbation, and its subsequent transition into the movement execution subspace results in a high velocity movement. Note that this hypothesis is consistent with the observation that preparatory activity in premotor and motor cortices, even while not resulting in execution, can be tuned to movement parameters, e.g., direction (*Churchland et al., 2010*). However, this correlation between preparatory activity and spatial metrics of the upcoming movement - which is also present in SC since the locus of activity determines where the movement is going to be directed - is not directly indicative of a motor potential. In our framework, movement-generating potential requires that the activity be correlated with concurrent movement dynamics, ensuring that

the activity actually makes its way down to the muscles. More studies that causally delink evolving population activity from physiological gating are needed to clarify these mechanisms.

## Implications for threshold-based accumulator models

The threshold hypothesis is the leading model of movement preparation and initiation in the gaze control system (*Hanes and Schall, 1996*). Inspired by stochastic accumulator models of decision-making (*Carpenter and Williams, 1995*; *Ratcliff and Rouder, 1998*), this theory posits that saccade initiation is controlled by accumulation to threshold of a motor preparation signal in premotor neurons. However, it is unclear whether the preparatory activity must rise to a fixed biophysical threshold at the level of individual neurons or a population of neurons in order to initiate a saccade (*Hanes and Schall, 1996*; *Zandbelt et al., 2014*), or whether there exists a dynamic equivalent to such a threshold (*Lo and Wang, 2006*; *Cisek et al., 2009*).

Another recent study has shown that the fixed threshold hypothesis does not hold true for most neurons in SC and FEF (*Jantz et al., 2013*), finding that the effective threshold varies based on the task being performed by the subject. However, comparison of thresholds across tasks is subject to the confound that the network may be in a different overall state, thereby modulating the threshold. For instance, the presence or absence of the fixation spot at the time of movement initiation, or the presence of other visual stimuli or distractors, may affect how downstream neurons receiving premotor activity from the whole network decode the level of activity, thus influencing the effective threshold. In our study, thresholds are compared between interleaved perturbation and control trials in the *same* behavioral paradigm, eliminating the effect of preset network-level confounds in the premotor circuitry within SC and upstream thereof.

Our results show that the low-frequency preparatory activity of individual or population of SC neurons does not have to transition into a high-frequency mode to trigger a movement (*Figure 6a, b*). If gates downstream of the SC are removed, then reduced SC activity is sufficient to move the eyes. The effective reduction in threshold (or equivalently, non-existence of a fixed threshold) that we observe is likely due to reduced activity of the OPNs, which are a potent source of inhibition on the pathway downstream of the SC. The OPNs are inhibited by the reflex blink, and thus premotor activity needs to overcome lesser inhibition and is able to trigger movements off a lower level. It is also important to note that while there is some evidence that premotor activity in SC is attenuated when saccades are perturbed by a reflex blink (*Goossens and Van Opstal, 2000a*), we did not observe suppression during movements that were triggered by the blink, as seen in the firing rate profile in *Figure 7a*. Nevertheless, the presence of any attenuation would only strengthen the result, since the occurrence of saccades despite attenuated SC activity goes against the notion of a rigid threshold.

## Related considerations

It has been hypothesized that the role of a threshold may perhaps be to arbitrate between speed and accuracy during decision making and movement planning (*Heitz and Schall, 2012*). Studies have provided contrasting evidence for this idea, with some suggesting a collapsing bound reflecting the urgency of a movement (*Cisek et al., 2009*; *Thura et al., 2012*), and others placing the onus of balancing speed with accuracy on non-threshold features of neural activity, such as baseline and gain (*Hanks et al., 2014*; *Salinas et al., 2014*). In our study, the accuracy of reduced latency blink-triggered movements was unaltered relative to normal saccades (*Figure 3b*), despite the effective reduction in threshold we observed. However, the accumulation rate increased following disinhibition by the blink, suggesting that threshold and rate may be independently modulated to achieve the requisite balance between speed and accuracy. In addition, the observation that reaction times can be reduced without affecting accuracy suggests that saccades may not be subject to a strict speed-accuracy tradeoff, at least in the context of a simple single target task such as the one used in this study. Strikingly, for such tasks, even higher accuracies can be observed in conjunction with faster responses if an appropriate behaviorally relevant parameter is manipulated (e.g., reward in *Takikawa et al., 2002*). Thus, while speed-accuracy tradeoffs have been implicated in optimal control of saccades when speed refers to saccade velocity (*Harris and Wolpert, 2006*), more complex tasks which invoke cognitive processes such as decision-making may be necessary to observe a tradeoff of accuracy with reaction speed.

Studies have shown that the latencies of saccades and hand movements are reduced under experimental conditions that introduce a startling stimulus (*Valls-Solé et al., 1995*), present targets with predictable spatial and temporal features (*Paré and Munoz, 1996*), or provide instructions to time the movement (*Haith et al., 2016*). We speculate that some combination of early onset of preparatory activity, higher baseline, or fast accumulation to a trial-specific threshold must be achieved before the movement is triggered. In terms of the population dynamics framework discussed a couple of sections back, the reduction in latency can be viewed as speeding up the transition of population signals from movement preparation to movement generation subspaces. It is important to note that the dynamics of movements evoked under the conditions used in these experiments are remarkably similar to control movements. In our view, reduced latency blink-triggered movements are not produced by the same neural mechanisms. We have reported previously that delivering an air-puff to the ear or neck does not reduce the latency of saccades, thus discounting the startle stimulus explanation (*Gandhi and Bonadonna, 2005*; *Jagadisan and Gandhi, 2016*). Here, we randomly interleaved blink and control trials, so preparatory signals cannot start to accumulate earlier and/or faster on perturbation trials. Finally, the dynamics of blink-triggered movements and control saccades are not similar. Our method thus reveals a motor potential component during the preparatory period that is not disclosed in the other studies.

## The role of SC and brainstem in saccade execution

Although it is well-known that the locus of activity on the SC map determines amplitude and direction of the saccade vector (see *Gandhi and Katnani, 2011*), for a review), it has thus far been unclear whether the instantaneous firing rate of SC neurons directly drives saccade kinematics, i.e., instantaneous velocity. In our view, the mini-vector model of saccade execution comes closest to specifying a direct relationship between SC spiking and saccade metrics (*Van Gisbergen et al., 1987*; *Arai et al., 1994*; *Goossens and Van Opstal, 2006*). This model proposes that each spike in SC contributes to a fixed length desired displacement of the eye during the window when gating is open, and has been tested with blink perturbations (*Goossens and Van Opstal, 2006*). But it is unclear how this translates to the motor potential we observe, either before or during the saccade, since we estimate this potential from correlated variability between SC activity and eye kinematics across trials, not within a trial. Moreover, in *Figure 5*, this correlation is computed between activity and residual velocity projected onto the direction of the saccade target after subtraction of the BREM template. This can cause the kinematic variable to be instantaneously negative (i.e., going away from the saccade target) on some trials, but as long as it is less negative on trials when the activity is higher, we can say that the activity has motor potential. In contrast, the mini-vector model will predict that the eye moves in fixed vector increments towards the saccade target (which happens roughly to be the optimal vector of the recorded neuron). Moreover, in the previous study, not all neurons show the fixed spike count property, and at the population level, spike counts on perturbation trials are slightly higher than on control trials. This is entirely consistent with our observation that spikes can leak through when the gating is open (leading sometimes to excess spikes as in *Goossens and Van Opstal, 2006*; also see *Buonocore et al., 2017*).

Previous work has shown that the discharge profiles of burst generator neurons in the reticular formation are a scaled version of the observed eye velocity waveforms (*Cullen and Guitton, 1997*). Moreover, in the presence of a perturbation - for example, when a natural blink accompanies a saccade (*Gandhi and Katnani, 2011b*) or when a brief torque is applied to the head during eye-head gaze shifts (*Sylvestre and Cullen, 2006*) - burst neuron activity is also modified to account for the observed changes in eye velocity. Given these results, we predict that the lower brainstem burst generator neurons will exhibit low frequency activity to produce the slow eye movement leaked by premature inhibition of OPNs, followed by a high frequency burst that generates the saccade.

We would like to emphasize that the observed results - preparatory motor potential, reduced threshold, accelerated activity dynamics - are most likely indirect effects of the trigeminal blink reflex, via inhibition of the OPNs, and not directly due to the reflex itself. Prior work has shown that the activity of SC neurons is not affected by the BREM produced during fixation (*Goossens and Van Opstal, 2000a*; *Jagadisan and Gandhi, 2016*). For blinks produced after the saccade target is presented, some SC neurons in fact exhibit attenuation (*Goossens and Van Opstal, 2000a*), although we did not see it in our dataset (and even if that did happen, it is counter-intuitive to and does not explain the motor potential and acceleration of activity). These observations collectively suggest that

the acceleration of activity that leads to a reduced latency saccade is not directly due to the trigeminal blink reflex but indirectly due to OPN inhibition.

In our study, we found the correlation between SC activity and eye kinematics during the saccade to be maximal at a time shift of 12 ms between the two signals, providing us with an estimate of the optimal efferent delay between SC and extraocular muscles. This is in line with previously estimated ranges for the efferent delay (8–17 ms, *Miyashita and Hikosaka, 1996*). In that study, single pulse microstimulation was delivered to the SC during an ongoing saccade, and the latency to deviation from normal saccade kinematics provided an estimate of the time it takes for a spike in SC to reach extraocular muscles while the gating in the pathway downstream of SC is open. The fact that we observe the same efferent delay (i.e., 12 ms) prior to saccade onset, when the OPNs are already quiescent due to the reflex blink, fits neatly within this picture and adds credibility to the notion of a latent motor potential in preparatory spikes.

## Parallel implementation of the sensory-to-motor transformation

An influential idea in systems neuroscience is the premotor theory of attention, which posits that spatial attention is manifested by the same neural circuitry that produces movements (*Rizzolatti et al., 1987*). Consistent with this hypothesis, studies with cleverly designed behavioral tasks have attributed low-frequency build-up of activity in premotor neurons to a number of cognitive processes, including target selection (*Schall and Hanes, 1993*; *Horwitz and Newsome, 1999*; *McPeek and Keller, 2002*; *Carello and Krauzlis, 2004*), attention (*Goldberg and Wurtz, 1972*; *Ignashchenkova et al., 2004*; *Thompson et al., 2005*), decision-making (*Newsome et al., 1989*; *Gold and Shadlen, 2000*; *Ramakrishnan and Murthy, 2013*), working memory (*Sommer and Wurtz, 2001*; *Balan and Ferrera, 2003*), and reward prediction (*Platt and Glimcher, 1999*; *Hikosaka et al., 2006*). However, such multiplexing of cognitive signals along with movement preparation and execution-related activity by premotor neurons only provides circumstantial evidence in support of the premotor theory, leaving open the question of whether the low-frequency activity exclusively represents cognitive and preparatory processes, devoid of the concurrent ability to generate a movement (a.k.a motor potential).

Efforts to delineate spatial attention and movement intention by means of causal manipulations have produced a mixed bag of results, with some studies supporting disjoint processing (*Juan et al., 2004*) and others supporting concurrent processing (*Moore and Armstrong, 2003*; *Katnani and Gandhi, 2013*). The strongest piece of evidence yet for concurrent processing is the observation that many of these premotor neurons also discharge following the onset of a visual stimulus (*Wurtz et al., 2001*), which can make its way down to effectors resulting in an electromyographic response, e.g., in the neck (*Corneil et al., 2004*). Such 'leakage' of decision-related activity down to the muscles has also been observed in other effectors, including reaches (*Corneil and Munoz, 2014*; *Servant et al., 2015*), and even across effectors (*Joo et al., 2016*). The discovery of a latent motor potential in the preparatory activity of SC neurons significantly advances this debate by suggesting that while the low-frequency build-up may not trigger movements under normal conditions, movement intention and motor programming signals are also encoded by these neurons in parallel. Moreover, unlike manipulations such as microstimulation or pharmacological inactivation that introduce extrinsic signals that may corrupt the natural processing of this activity (*Katnani and Gandhi, 2013*), reflex blinks are non-invasive and are likely to provide a more veridical readout of ongoing processes.

## Concluding remarks

It is worthwhile to end on a note of caution. The results in this study are based on experiments performed in one node, SC, in a distributed network of brain regions involved in gaze control. Traditional knowledge imposes a hierarchy on the sensorimotor transformations that need to occur before a gaze shift is generated (*Wurtz et al., 2001*). It is possible that sensorimotor neurons in SC, and to some extent, FEF, which project directly to the brainstem burst generator (*Segraves, 1992*; *Rodgers et al., 2006*), are more likely to exhibit signatures of a motor potential in preparatory activity compared to regions higher in the cascade. Neurons in other regions may still need to signal the initiation of a movement by reaching a threshold, optimal subspace, or other similar decision bound. Furthermore, it is known that movement initiation thresholds in SC and FEF can vary based on task

context (e.g., *Jantz et al., 2013*). The results presented here are based on a relatively simple task – the delayed saccade task. The mechanisms of movement initiation, and the presence of motor potential in preparatory activity, could in principle be different in more complex tasks, e.g. those that involve competitive spatial selection of movements or sequential movements. Future studies that take causal approaches to perturbing intrinsic population dynamics in various premotor areas across different tasks and effector modalities are essential in order to gauge whether the findings in this study point to a fundamental and generalizable property of sensorimotor systems.

## Materials and methods

### General and surgical procedures

All experimental and surgical procedures were approved by the Institutional Animal Care and Use Committee at the University of Pittsburgh and were in compliance with the US Public Health Service policy on the humane care and use of laboratory animals. We used two adult rhesus monkeys (*Macaca mulatta*, 1 male and 1 female, ages 8 and 10, respectively) for our experiments. Under iso-flurane anesthesia, a craniotomy that allowed access to the SC was performed and a recording chamber was secured to the skull over the craniotomy. In addition, posts for head restraint and scleral search coils to track gaze shifts were implanted. Post-recovery, the animal was trained to perform standard eye movement tasks for a liquid reward.

### Visual stimuli and behavior

Visual stimuli were displayed by back-projection onto a hemispherical dome. Stimuli were white squares on a dark grey background, 4 × 4 pixels in size and subtended approximately 0.5° of visual angle. Eye position was recorded using the scleral search coil technique, sampled at 1 kHz. Stimulus presentation and the animal's behavior were under real-time control with a LabVIEW-based controller interface (*Bryant and Gandhi, 2005*). After initial training and acclimatization, the monkeys were trained to perform a delayed saccade task. The subject was required to initiate the trial by looking at a central fixation target. Next, a target appeared in the periphery but the fixation point remained illuminated for a variable 500–1200 ms, and the animal was required to delay saccade onset until the fixation point was extinguished (GO cue). Trials in which fixation was broken before peripheral target onset were removed from further analyses. The animals performed the task correctly on >95% of the trials. During each session, we presented the targets in one of two locations – either inside the response field of the recorded neuron, or in the diametrically opposite location (see below).

### Induction of reflex blinks

On 15–20% of trials, fixation was perturbed by delivering an air puff to the animal's eye to invoke the trigeminal blink reflex. Compressed air was fed through a pressure valve and air flow was monitored with a flow meter. To record blinks, we taped a small Teflon-coated stainless steel coil (similar to the ones used for eye tracking, but smaller in coil diameter) to the top of the eyelid. The air pressure was titrated during each session to evoke a single blink. Trials in which the animal blinked excessively or did not blink were excluded from further analyses. The air-puff delivery was randomly timed to evoke a blink either during fixation (400–100 ms before target onset) or 100–250 ms after the GO cue, during the early phase of the typical saccade reaction time. The blink evoked during fixation produced a slow and loopy blink-related eye movement (BREM, gray traces in *Figure 2a*). The eyes returned to near the original position and fixation was re-established for 400–100 ms before a target was presented in the periphery and the remainder of the delayed saccade task continued. The window constraints for gaze were relaxed for a period of 200–500 ms following delivery of the air puff to ensure that the excursion of the BREM did not lead to an aborted trial. The blink evoked after the GO cue typically produced a blink-triggered movement that can be described as a combination of a BREM and a saccade to the desired target location (colored traces in *Figure 2a*). We used deviations from the BREM profile to determine true saccade onset, as described in more detail in the next section.

## Movement detection

Data were analyzed using a combination of in-house software and Matlab. Eye position signals were smoothed with a phase-neutral filter and differentiated to obtain velocity traces. Normal saccades, BREMs, and blink-triggered gaze shifts were detected using standard onset and offset velocity criteria (50 deg/s and 30 deg/s, respectively). Onsets and offsets were detected separately for horizontal and vertical components of the movements and the minimum (maximum) of the two values was taken to be the actual onset (offset). Saccade onset times within blink-triggered movements were detected using a non-parametric approach (*Katnani and Gandhi, 2013*, also see *Figure 2a*). We first created an estimate of the expected BREM distribution during each session by computing the instantaneous mean and standard deviation of the horizontal and vertical BREM velocity profiles. Then, for each blink-triggered movement in that session, we determined the time point at which the velocity exceeded the ±2.5 s.d. bounds of the BREM profile distribution, and remained outside those bounds for at least 15 successive time points. We did this separately for the horizontal and vertical velocity profiles, and took the earlier time point between the two components as the onset of the saccade. We further manually verified that the detected saccadic deviations on individual trials were reasonable, esp., in the spatial domain. *Figure 2a* illustrates this approach for three example trials.

## Electrophysiology

During each recording session, a tungsten microelectrode was lowered into the SC chamber using a hydraulic microdrive. Neural activity was amplified and band-pass filtered between 200 Hz and 5 kHz and fed to a digital oscilloscope for visualization and spike discrimination. A window discriminator was used to threshold and trigger spikes online, and the corresponding spike times were recorded. The location of the electrode in the intermediate layers of SC was confirmed by the presence of visual and movement-related activity as well as the ability to evoke fixed vector saccadic eye movements at low stimulation currents (20–40 µA, 400 Hz, 100 ms). Before beginning data collection for a given neuron, its response field was roughly estimated. During data collection, the saccade target was placed either in the neuron's response field or at the diametrically opposite location (reflected across both axes) in a randomly interleaved manner. Response field centers, and therefore, target locations (also consequently, saccade amplitudes and directions) varied between 9–25 degrees in eccentricity and spanned all directions.

## Data analysis – neural pre-processing

Spike density waveforms were computed for each neuron and each trial by convolving the raw spike trains with a Gaussian kernel. We used a 3 ms wide kernel for the motor potential and threshold analysis (involving across-trial correlations or trial-averaged neural activity) and a 10 ms kernel for the accumulation rate analysis (for better rate estimation on individual trials). For a given neuron and target location, spike densities were averaged across trials after aligning to target and saccade onsets. Neurons were classified as visual, visuomovement or movement-related, based on the presence of a significant target and/or saccade-related response. We only analyzed visuomovement and movement neurons for this study, the majority of which were visuomovement (47/50). Where necessary, we normalized the trial-averaged spike density of each neuron to enable meaningful averaging across the population. The activity of each neuron was normalized by its peak trial-averaged firing rate during normal saccades.

## Data analysis – inclusion criteria

Overall, we recorded from 64 neurons for 12339 control trials and 2364 blink trials over 50 sessions. For all analyses, we only considered neurons for which we had at least 7 blink perturbation trials with the target in the response field. Since we only introduced the blink perturbation on a small percentage of trials in a given session in order to prevent habituation, this restricted our population to 50 neurons (7891 control trials and 1615 blink trials over 43 sessions). We used all 50 neurons for the threshold analysis (*Figure 6*). For the motor potential analysis (*Figure 5*), since our aim was to correlate neural activity with eye kinematics before saccade onset, we used only the subset of trials where the onset of the saccade was delayed with respect to overall movement onset by at least 20 ms (see *Figure 2b*). To ensure that the correlation values were reliable, we used only neurons which had at

least 7 trials meeting the above criterion. This restriction reduced the number of neurons available for the motor potential analyses to 38 (6771 control trials and 869 blink trials over 32 sessions), and we used the same neurons for control trials to enable meaningful comparison (*Figure 4*). For the same reason, we also used this subset of neurons for the accumulation rate analysis (*Figure 7*), where we compared the dynamics of neural activity in 20 ms windows before and after blink onset, and we wanted to ensure that the post-blink window did not include activity co-occurring with the saccade.

## Kinematic variables

For the motor potential analyses in *Figures 4* and *5*, we computed the across-trial correlation between instantaneous movement kinematics and neural activity for each neuron. We computed the kinematic variable of relevance for each analysis as follows. In all cases, we used the raw or modified (see below) horizontal and vertical velocity signals to compute a single vectorial velocity signal using the Pythagorean theorem: $v(t) = \sqrt{v_h^2(t) + v_v^2(t)}$. For the analyses in *Figure 4—figure supplement 1*, for control trials, we used the raw, unmodified velocity signals to compute vectorial velocity as a function of time, which we then used as the instantaneous kinematic variable to correlate with neural activity. For perturbation trials in *Figure 5* and *Figure 5—figure supplement 1*, we first noted that the blink-related eye movement (BREM) contributes a substantial velocity component to the overall movement, since the initial phases of velocity and spatial profiles of blink-triggered movements look largely like those of a BREM (see *Figure 2a*). Thus, in order to extract only the saccadic component of a blink-triggered movement, we subtracted from it the mean BREM template on a given session, and used only the horizontal and vertical residuals to compute the vectorial residual velocity: $\tilde{v}(t) = \sqrt{\tilde{v}_h^2(t) + \tilde{v}_v^2(t)}$, which was used for the correlation analysis in *Figure 5—figure supplement 1*.

A potential pitfall when using residual velocities by just subtracting out the mean BREM, given the variability in BREM profiles across repetitions, is that intrinsic variability of the BREM itself may mask any correlated variability that might be present between ocular kinematics and neural activity. In other words, if the BREM is driven by an independent pathway compared to the saccade/SC activity, it represents an orthogonal source of variability in the kinematics relative to the activity-driven variability that is being examined. Therefore, for the perturbation trial analysis in *Figure 5*, we used the component of residual velocity in the direction of the saccade goal, to isolate variability in the direction of the saccade. The kinematic variable for this analysis is thus defined as: $\tilde{v}_\theta(t) = \sqrt{\tilde{v}_h^2(t) + \tilde{v}_v^2(t)} \cos\theta$, where $\theta$ is the angle between the instantaneous residual velocity vector and the direction of the saccade goal (e.g., between the green and black vectors in the inset in *Figure 5a*). For the sake of consistency, we used a similar variable: $v_\theta(t) = \sqrt{v_h^2(t) + v_v^2(t)} \cos\theta$, for the equivalent control analysis in *Figure 4*, even though the instantaneous direction of velocity is largely towards saccade goal in this condition.

## Motor potential estimation

To estimate motor potential, we computed the correlation between instantaneous neural activity and eye kinematics (according to the variables defined above) across trials, across a span of efferent delays (time shifts) between activity and velocity. For each neuron, we computed the Pearson's correlation coefficient $c_{t+\Delta, t} = corr(\boldsymbol{a}(t+\Delta), \boldsymbol{v}(t))$ between the activity vector $\boldsymbol{a}(t) = [a^1(t), \ldots, a^n(t)]$ and the kinematics vector $\boldsymbol{v}(t) = [v^1(t), \ldots, v^n(t)]$ at time separations or lags $\Delta \in [-50, 50]$, where n is the number of trials for that condition for that neuron. Each point in panel c in *Figures 4–5* and associated figure supplements 1 represents the average correlation coefficient across the population of neurons between firing rate and kinematics at the corresponding time points $(t, t+\Delta)$. To estimate the optimal efferent delay between activity and velocity, we computed the time at which this population average correlation peaks along the activity axis (vertical axis in the panel c heatmaps) for each time point during the movement. A moving average (5 ms window) of this efferent delay trace is shown as the black trace in the heatmaps. The vertical distance of this trace from the unity line is equal to the the actual value of the optimal efferent delay at each movement time point, shown in panel d in *Figures 4–5* and associated figure supplements 1. We then calculated the mean efferent delay during the movement after saccade onset (and before saccade onset in the perturbation

condition), and plotted the population average correlation at this delay (panel e in the aforementioned figures).

## Surrogate data and statistics

For the accumulation rate analysis in *Figure 7*, we created a surrogate dataset of control trials with blink times randomly sampled from the distribution of blink occurrences in perturbation trials for that session and assigned to individual control trials. For each neuron, we created 1000 such pseudo-trials by resampling from and reassigning to control trials. We then fit the accumulation rates 20 ms before and after the blink with piecewise-linear functions. The slopes of the linear fits were taken to be the pre- and post-blink rates of accumulation for each neuron. We then compared the change in accumulation rates before and after the pseudo-blink in control trials and the blink in perturbation trials by computing the rate modulation index for each condition as $\frac{rate_{post} - rate_{pre}}{rate_{post} + rate_{pre}}$. Note that we also created a similar surrogate dataset for control trials for the purpose of visualization alone in *Figure 3a*.

To estimate the significance of the correlation profile in the motor potential analyses, we performed a bootstrap analysis on a trial-shuffled dataset using the motor potential estimation procedures laid out in the previous section. For each neuron, we shuffled trial identities between the across-trial activity and velocity vectors, and computed population average correlation as before. We performed this for 100 different shuffles, and the resultant across-shuffle mean correlation trace at the optimal efferent delay (as estimated from the true data) and the 95% confidence interval bounds are shown in panel e in *Figures 4–5* and associated figure supplements 1. At each time point, we calculated the difference between the actual correlation profile and each iteration of the bootstrap, and computed the 95% confidence interval of this difference distribution. The correlation at a particular time was considered significant if this interval did not contain 0. For comparisons of threshold and accumulation rate between control and blink-triggered conditions, we used appropriate non-parametric statistical tests – the Wilcoxon rank sum test when comparing distributions across trials, and the Wilcoxon signed-rank test for trial-averaged comparison across a population.

## Additional information

### Funding

| Funder | Grant reference number | Author |
| --- | --- | --- |
| National Eye Institute | EY022854 | Neeraj J Gandhi |
| National Eye Institute | EY024831 | Neeraj J Gandhi |

The funders had no role in study design, data collection and interpretation, or the decision to submit the work for publication.

### Author contributions

Uday K Jagadisan, Conceptualization, Data curation, Software, Formal analysis, Investigation, Visualization, Writing—original draft, Writing—review and editing; Neeraj J Gandhi, Conceptualization, Resources, Supervision, Funding acquisition, Visualization, Methodology, Project administration, Writing—review and editing

### Author ORCIDs

Uday K Jagadisan (iD) http://orcid.org/0000-0003-4253-6041
Neeraj J Gandhi (iD) http://orcid.org/0000-0002-4915-2131

### Ethics

Animal experimentation: This study was performed in strict accordance with the recommendations in the Guide for the Care and Use of Laboratory Animals of the National Institutes of Health. All of the animals were handled according to approved institutional animal care and use committee (IACUC) protocols of the University of Pittsburgh (Protocol 14114861).

Decision letter and Author response
Decision letter https://doi.org/10.7554/eLife.29648.017
Author response https://doi.org/10.7554/eLife.29648.018

## Additional files

**Supplementary files**
• Transparent reporting form
DOI: https://doi.org/10.7554/eLife.29648.013

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
