## [Decision Letter]

[Editors’ note: a previous version of this study was rejected after peer review, but the authors submitted for reconsideration. The first decision letter after peer review is shown below.]

Thank you for submitting your work entitled "Removal of inhibition uncovers latent movement preparation dynamics" for consideration by *eLife*. Your article has been reviewed by three peer reviewers, and the evaluation has been overseen by a Reviewing Editor and a Senior Editor. The reviewers have opted to remain anonymous.

Our decision has been reached after consultation between the reviewers. Based on these discussions and the individual reviews below, we regret to inform you that your work will not be considered further for publication in *eLife*.

All reviewers were generally quite positive about the ideas put forward in the study, and that it could potentially make a relevant contribution to the field. But all three reviewers also had several major concerns, which were broadly similar across the reviewers. Each reviewer voiced additional ones that were mostly distinct.

On consultation among the three reviewers, there was a consensus that these concerns are of sufficient magnitude and scope that the manuscript should be rejected, and the main points from this consultation are summarized below, in addition to the full reviews.

Briefly, these concerns can be summarized as follows: presently, the overall clarity of the research question and theory requires improvement and streamlining, and a more coherent argument how each of the results addresses the questions at hand. Additionally, the overall detail of methods and results could be improved, with important information not provided or unclear, which makes the assessment of parts of the results difficult. All three reviewers felt that the threshold aspect of the paper is the least interesting, and that the key interesting finding appears to be in Figure 7. This finding however relates to the idea of motor potential, and whether this is an intrinsic property of SC activity or whether motor potential of SC activity is determined downstream (by presence/absence of OPN activity). All reviewers felt that clarity might be improved by focusing on the question of motor potential, with the threshold being the putative mechanism by which SC activity could conceivably undergo a qualitative change in its capacity to elicit movement.

Overall, the analyses do not seem to fully align with the conclusions drawn from them. The number of neurons included is small, with considerable variation in firing rates, and further, more substantial analyses on how activity relates to pre–saccade activity may require additional recordings. A related concern is about the key analyses that seek to establish the 'motor potential' of pre–saccadic SC activity through a correlation between activity and eye velocity. The authors acknowledge the challenge in this. The results rest on being able to determine when the saccade begins, and the reviewers raise concerns that this was indeed possible with the precision required in the subsequent correlations with neural activity. Reviewer 3 points out that eye velocities could establish which eye movements were pre–saccadic, but e.g. a lack of reporting of saccade size makes it difficult to judge whether this might work. The current set of analyses does not seem to convincingly demonstrate that pre–saccadic SC activity does in fact influence eye kinematics during BREMs.

We also attach some general comments that came from the consultation, that you may find helpful for revising your paper, and that clarify some of the reviewers’ original comments:

A conceptual figure explaining the hypotheses, including a basic model of how SC, OPNs and burst generators interact, might be helpful for the reader.

Reviewer 1 asks about neuron properties in SC, but acknowledges not being an expert and this has therefore not directly impacted our assessment. The issue was indeed clarified by reviewer 3 who has experience in neural recordings. Reviewer 3 notes that variability such as reported here can be seen in the SC, and can strongly depend on different sized saccades and different directions of saccades. There are also classes of SC cells that show a strong motor burst only during the movement (with a very small temporal offset from movement onset), and maybe these will have less variability and show a fixed threshold. However, the analyses and details provided do not allow for assessing in more detail how such factors may have contributed to the results, and the small number of neurons seems to make it unlikely this can be addressed without additional data.

There are important links with other work, especially Kaufmann and colleagues and the work of van Opstal. Kaufmann, (2014) argue that 'motor potential' is a property intrinsic to motor cortical activity – opposite of the conclusion here about the SC. A key distinction between saccades and reaches might be that the trigger for reaches is 'upstream', perturbing the motor plan into potency, whereas for saccades, the trigger is implemented downstream, via the OPNs. Van Opstal and colleagues have suggested that each spike in the SC can contribute to saccade kinematics, but only consider spikes during movement. An additional "gating" mechanism is thus invoked to explain that many spikes occur outside saccades and without eye movements. The observation in your Figure 7 that spikes outside a saccade may contribute to movement properties seems relevant in this regard, but the analyses in this figure aren't supporting this as clearly as suggested.

Reviewer #1:

Summary:

This study tests the influential threshold account for saccadic movements. The main finding is that SC neurons do not need to reach a fixed activity threshold to trigger a movement. The authors derive this from showing a latent motor potential in the preparatory activity of SC neurons, which they uncover by removing the inhibitory influence on SC neurons through blink–triggered saccades.

The study is interesting, and speaks to a long–standing debate about the neural mechanism underpinning movement planning and initiation. The idea that activity has to reach a fixed threshold has gained a lot of attraction in the field, and this study questions that such a fixed threshold is indeed required. The use of blink–triggered saccades as a perturbation as used here seems elegant.

My enthusiasm is somewhat dampened by the complexity of the paper and by being unsure about some of the effects and the interpretation.

Firstly, if I'm correct, the authors seek causal evidence for movement occurring below threshold, by disinhibiting saccade activation via BREMs? Is this specific to the preparatory process? At one point the authors set out to test "how the brain prepares for movements before issuing the command to initiate them". This seems different from asking whether preparation and execution might occur in parallel or not, which the abstract's final sentence alludes to. This seems relevant to judge whether the data support the conclusions the authors make, which alternate between the threshold theory, and preparatory activity influences.

Second, there does not seem a behavioural manifestation of preparation. Was there any?

Third, are the two possibilities outlined in the Introduction indeed mutually exclusive? Could a parallel system not be equipped with a threshold mechanism, whereby there is continuous and parallel influence of preparatory signals, but even then, there is an overall activity level that will unleash the movement? The threshold theory states that a movement is triggered when a specific threshold is reached – at least in the standard cases usually tested. But it does not say that activity prior does not possess motor potential. A sprinter would normally start running when a go signal occurs, but he could start at any time prior to that. There is simply a good reason to inhibit that tendency because he would get disqualified otherwise. Does a threshold perhaps arbitrate between speed and accuracy?

Fourth, I'd like to be assured that the results indeed support the conclusions. There is some under–reporting of procedures, but it is also hard to gauge some of the neural data given the different transformation conducted. I list some of these under minor comments.

One point I'd like to clarify, noting that I am not an expert on SC firing rate properties. The key results are reported over a very short time window of e.g. 20ms. How then can spike rates per second of 400 or more be inferred robustly? The authors use a convolution approach but it would be reassuring to see raw firing rates here, and to be convinced that the results do not contain any spurious values from the convolution. But even if this is not the case, out of the neurons reported some have spikes/s of 10–20, others of 400+. Is this within the normal range of firing rates in SC neurons coding for saccades? And even if, I wonder how with this range all neurons can indeed relate to velocity so closely?

Reviewer #2:

In this paper, Jadagisan and Gandhi argue against the old but popular notion that movement initiation occurs as a consequence of preparatory activity exceeding a fixed threshold. Focusing on saccade preparation in the superior colliculus, they evoke saccades with lower–than–normal reaction times by eliciting the trigeminal blink reflex – a response known to lead to inhibition of omnipause neurons (OPNs) that inhibit eye movement. SC activity prior to these blink–evoked saccades is clearly lower than prior to normal saccades, contrary to a putative fixed threshold. Furthermore, activity prior to saccade onset appears to influence the kinematics of eye movements during the blink, suggesting that the preparatory activity has the potential to directly influence movement if allowed to via inhibition of the OPNs.

I am sympathetic to many of the ideas in this paper. However, I have reservations about the strength of the results and conclusions. First, refuting a fixed threshold for SC activity to elicit a saccade does not seem like a surprising result, particularly in light of papers cited by the authors which also claim that the threshold may be flexible or task–dependent. The more novel claim is therefore that pre–saccadic activity has 'motor potential' that can be unmasked by the inhibition of OPN activity during the trigeminal blink reflex. However, I have some significant concerns as to the veracity of this result.

'No threshold' versus 'Flexible threshold'

A central claim of the paper is refuting the notion that SC activity must reach a specific threshold in order to trigger a saccade. However, the manuscript is inconsistent on whether it purports to demonstrate the absence of a threshold, or simply a flexible one. The rhetoric seems to frequently switch between these two positions. For instance, the abstract claims that "the results bring into question threshold models for saccade generation" while the significance statement states that "the threshold is not fixed, but can be flexibly modulated based on the level of inhibition". This incoherence persists throughout the manuscript (e.g. subsection "Blink–triggered saccades are evoked at lower thresholds compared to normal saccades" vs "it is NOT necessary for activity in SC to reach a threshold level in order to produce a movement"). I don't believe the data can disambiguate between a flexible threshold and no threshold.

Evidence of 'motor potential'

Given that the notion of a flexible threshold is not altogether novel (albeit it is a nice, clear demonstration), the claim that 'sub–threshold' SC activity can influence movement becomes critical to the paper.

The authors establish the 'motor potential' of pre–saccadic SC activity through a correlation between activity and eye velocity. A major problem with this approach, however, as the authors acknowledge, is that in order to determine which activity is pre–saccadic, they need to identify when the saccade begins, but this is difficult to determine precisely when there is also a blink–related eye movement. It does not seem unlikely that in certain trials, the 'true' saccade onset time might have been earlier than estimated, in which case it would not be surprising for the "pre–saccade" activity to correlate with "pre–saccade" eye velocity (since both are actually post–saccade–onset).

A related issue is that saccade onset is determined based on when eye velocity exceeds a threshold z–score (based on the distribution of blink–related eye movements). Given the smoothness of the eye movements, the eye velocity in the 20ms preceding the detected onset of the saccade is bound to be correlated with the peak eye velocity. (e.g. see Figure 7A, lower panel, top line; because the saccade has negative velocity, it is inevitable that eye velocity will be negative as it breaks the initiation–detection threshold). So then the correlation between 'pre–saccade' activity (which will be correlated with activity just after the saccade starts) and 'pre–saccade' eye velocity becomes inevitable.

To guard against these problems, the authors analyze a window of activity even further from saccade onset ([–40,–20]ms, as opposed to [–20,0]ms). The problems with analyzing this earlier window are that (i) the correlations between activity and velocity are extremely weak, (ii) for the example neuron, the eye velocity associated with SC activity seems to be negative in most trials, hardly demonstrating 'motor potential', and (iii) it is by no means clear that moving the window forward by 20ms alleviates the confounds described above (i.e. the faint correlations may still be driven by accidentally catching peri–saccade activity in enough trials, or because of smoothness of the eye velocity signal as it crosses threshold). I believe the authors need to provide a much stronger argument than that that their primary result is not undermined by these potential confounds.

*Reviewer #3:*

The authors used a blink perturbation to shift visually guided saccades towards having lower latencies than usual. This allowed measuring SC activity for these "premature" saccades. SC neurons burst for the premature saccades, but the saccades were elicited at lower firing rates. Also, the low–level activity in the SC "leaked" into downstream structures with the perturbation, and affected eye movement trajectories. The authors discussed these results in relation to rise–to–threshold models of saccade triggering, arguing that a fixed threshold is not necessary for movement generation. The authors interpreted the "leakage" as the pre–saccadic building possessing a motor potential.

The paper is generally quite well written (particularly the introduction), but there are notable important deficiencies in clarity in several places highlighted in the more detailed comments below. I would strongly suggest that the authors provide more details in general throughout the manuscript (e.g. neuron depths, locations in the SC topography map, and saccade sizes and directions etc).

In terms of the study itself, I found that the results of Figures 6 and 7 are the most important and most interesting aspects of this study. I would highly recommend that the authors expand on these results even more, if possible (even at the expense of the earlier results, which are a bit less exciting). For example, do these correlations get modulated with rostro–caudal position of the neurons? How do these observations relate to the "mini–vector" models of Van Opstal? What prevents "leakage" without the perturbation? etc. The description felt a bit light even though I find this result much more interesting than the threshold stuff.

Another question is how the threshold results relate to pure saccade bursters. What would their burst size look like for the short latency saccades after the blink manipulation? And, if it is not affected, then what does this imply for their interpretation?

– Introduction: I think I understand what you mean, but the way things are phrased, it might appear to contradict the abstract. In other words, in the abstract, you say that the preparatory activity has motor potential. Here, in the paragraph of the Introduction, you describe the two possibilities: (1) parallel, (2) serial. In (1), the preparatory activity would have the "motor potential" as you say in the abstract. In (2), the potential is acquired only later. So, my problem comes when you say "…presence of *such* a motor potential…". Given the positioning of the definition of motor potential in the previous paragraph, the use of "such" here makes the reader think that you will show in your results scenario (2) above instead of scenario (1), which would contradict the Abstract. I would rewrite the sentence “How might we test for the presence of such a “motor potential” in low frequency neural activity? “to be more explicit and avoid confusion of a possible contradiction with the Abstract (expand the sentence and remove "such").

– Introduction and Figure 7: I think these results are the most interesting aspect of this study.

– Results, I think for a broad journal like this one, "the delayed saccade task" is way too brief a description. Please explain the task briefly here (even if you do it later in the methods section, which you actually did not do sufficiently either). Statements like "after the go cue" would be foreign to a great majority of readers.

– Figure 2 legend, please define BREM even if it was defined in the text. Some readers quickly parse through figures before reading the text, and they would not know what BREM means.

– Figure 2 legend: do you mean "relative to movement onset" or "relative to blink onset"? Similarly, for "after movement onset". I assume you mean "after blink onset". Please clarify. This is very confusing especially because there are different types of "movement" in your paradigm.

– Figure 2: the colours in panel B are not defined in the figure legend (and very late in the main text – also, in the text, the justification for the division into the two colours is not clear at all).

– Results: what's the point of Figure 2B? I think that readers should be walked through the logic of your manipulation and why you feel the need to show Figure 2B, and why there are two different colours etc. Also, the key is really Figure 3, and both Figure 2 and 3 are more introductory and confirmatory than anything else. I would suggest combining Figures 2 and 3 (as one related methods figure) and explaining clearly in the text how your manipulation leads to your goal of getting short RT's. This is an example of a strong deficiency in clarity of the text.

– Subsection “The blink perturbation triggers reduced latency saccades”: what does "with saccade onset greater than 20ms" mean? This part of the text is very unclear.

– Subsection “The blink perturbation triggers reduced latency saccades”: normalized to what?

– Figure 4B: please indicate in the legend that the n=50 is the number of neurons, but that you show more points in the figure My understanding is that you show 50 points for each time window highlighted in A, correct?

– Figure 6, please indicate the sizes and directions of the saccades, and especially in panel A.

– Subsection “SC preparatory activity preceding the saccade-related burst possesses motor potential”: pointing to methods "for more details" seems insufficient here. Explain to the readers more especially because this is critical for the most important aspect of the study.

– Subsection “SC preparatory activity preceding the saccade-related burst possesses motor potential”: again, similar idea. why 15ms? etc

– Subsection “SC preparatory activity preceding the saccade-related burst possesses motor potential” needs more clarification

– Description of Figures 6–7: I didn't see the p–values associated with all these correlations. Please add them.

– Subsection “Implications for threshold-based accumulator models”: is it established that OPN's implement strict gating?

– Materials and methods and paper in general: there are a lot of missing details. Where did you record in the SC both in terms of rostro–caudal/medio–lateral position and also in terms of depth? How large were the saccades? What were their directions? Is there dependence of the effects on all of these SC and saccade parameters?

[Editors’ note: what now follows is the decision letter after the authors submitted for further consideration.]

Thank you for resubmitting your work entitled "Removal of inhibition uncovers latent movement potential during preparation" for further consideration at *eLife*. Your revised article has been favorably evaluated by Sabine Kastner (Senior editor) and three reviewers, one of whom is a member of our Board of Reviewing Editors.

The manuscript has been substantially improved, and all reviewers thought the work constitutes a valuable contribution. However, there are some remaining issues that need to be addressed before acceptance, as outlined below:

Reviewer #1:

This is an interesting, careful study looking at responses in the superior colliculus (SC) and their contribution to saccade initiation and to the metrics of the evoked eye movements. The use of blink–triggered saccades provides a valuable tool for manipulating the dynamics of the oculomotor circuitry in a relatively natural yet informative way. The results provide important evidence showing that the subthreshold, preparatory activity typically seen before saccades partially specifies the motor command itself, as reflected in saccade kinematics. Furthermore, they demonstrate that the threshold for triggering a saccade is, at the very least, more flexible than is typically assumed. The findings are significant because the concept of a fixed threshold has become dogma in studies of decision making, yet relatively little is known about it in neurophysiological terms, even for the clearest instance – precisely that of saccade generation.

My only criticism is that the way the conclusion about the threshold is articulated is a bit too narrow; perhaps misleading. To begin with, it is unclear what or where exactly the threshold *is*, in neural terms, so the assertion that "reaching threshold is not a necessary condition for movement initiation" (Abstract, Introduction, subsection “Blink-triggered saccades are evoked at lower thresholds compared to normal saccades”) seems extreme, and at some level contradictory. For instance, the blink might simply decrease the required threshold level. Also, the term "threshold" in a way describes the ballistic nature of the saccade; one could argue that, empirically, the threshold is simply whatever happens that turns the current motor activity into an uncancellable motor command – but it is unclear what exactly this is, or whether *the* threshold is really a collective property to which, say, LIP, FEF, and SC all contribute. What is clear is that the threshold is not fixed, and that threshold crossing may have something to do with inhibition of the OPNs, as the authors already imply. The Discussion shows that the authors appreciate the difference between existence and flexibility, but I think rephrasing this main conclusion in a more nuanced way at the various points at which it is mentioned would improve the clarity of the main message.

Reviewer #2:

Overall, the authors have done an outstanding job revising the manuscript. The manuscript now has a much clearer conceptual focus, and more rigorous and convincing analysis. I feel this is now a strong and compelling paper. I have no major outstanding concerns.

Reviewer #3:

I like this revised version much. It's very nice. It is clear for the most part, and it has addressed the previous comments. I have only a few suggestions for improvements:

– Results section: some parts are difficult to understand. I would explain the divide between the <20 and >20ms movements better. More importantly, later in the same set of lines when you refer to "motor potential analyses" and the "threshold analyses", these two concepts are a bit foreign still, and just saying "the threshold analysis" or "the motor potential analysis" assumes that readers know what "the" analyses are. I know that you alluded to them earlier in the introduction, but it still helps to rewrite this final sentence in a more expositive way especially so early in the paper. For example, perhaps you can say something along these lines: because blink–triggered movements with saccade onset >20ms after blink onset likely reflect the idea that they were triggered anew by the blink–related disinhibition, they would be ideal for exploring whether there is a motor potential in the period even well before the actual saccade onset. On the other hand, for movements with <20ms, these are movements for which motor preparation was well under way, and are thus ideal to explore whether triggering can happen even before a fixed threshold is reached. (In other words, explain a bit more verbally the different "analyses" that will come up later in the paper. My exact wording above is probably bad but you can come up with the perfect wording).

– The motivation for Figure 3 is still somewhat unclear to me. Doesn't it just replicate previous results? If so, then why show it as a main figure? Perhaps it could be included as a supplementary figure at an appropriate time later when you want to address a possible confound in interpreting motor potentials or what have you? It's not entirely clear why the reader needs to go through a new replication of older results from the same lab. I guess the point to make is in the Results but if this point can be made with citations, then this is just fine.

– Subsection “SC preparatory activity preceding the saccade-related burst possesses motor potential”: better say "after time 0 in the x–axis in the heat map in Figure 5C" because both x and y axes have time in them, so it's better to clarify which "time 0" you mean.

– Please explain Figure 5B in more detail in the text. It is not described at all in the text, but the reader would want to know the difference from Figure 4B. So, just walk us through a textual description of Figure 5B and what it means for your interpretation and when comparing to Figure 4B. What are we supposed to understand from looking at this figure? This would also be a nice segway into describing Figure 5C. Currently, I'm not sure what the take home message from Figure 5B is based on reading just the results text.

– In the section of Figure 5, you don't mention different rostra–caudal positions. I know that in the responses document, you say that there was a negative result, but this needs to be mentioned in the Results section and then maybe discussed briefly in the discussion. For example, you argue that the target was always the optimal target for the neuron. However, different neurons along the rostra–caudal extent of the SC will have different projection strengths to brainstem pre–motor neurons, right? So, does the speed of the pre–saccadic eye movement in blink triggered movements scale with the target eccentricity? This would lend strong support to your arguments in subsection “SC preparatory activity preceding the saccade-related burst possesses motor potential”.

– Subsection “Blink-triggered saccades are evoked at lower thresholds compared to normal saccades”: perhaps it could be nice to start with a summary sentence: “Our results so far show that… and…”. Then, you can say: “next, we used the fact… to study a related issue of what it takes to trigger the movement itself."

– Subsection “Blink-triggered saccades are evoked at lower thresholds compared to normal saccades”: here could be a place to address saccade accuracy (Figure 3) as a supplementary figure. For example, you can say that they are triggered earlier and with lower activity, even though they are still spatially accurate.

---

## [Author Response]

Briefly, these concerns can be summarized as follows: presently, the overall clarity of the research question and theory requires improvement and streamlining, and a more coherent argument how each of the results addresses the questions at hand. Additionally, the overall detail of methods and results could be improved, with important information not provided or unclear, which makes the assessment of parts of the results difficult. All three reviewers felt that the threshold aspect of the paper is the least interesting, and that the key interesting finding appears to be in Figure 7. This finding however relates to the idea of motor potential, and whether this is an intrinsic property of SC activity or whether motor potential of SC activity is determined downstream (by presence/absence of OPN activity). All reviewers felt that clarity might be improved by focusing on the question of motor potential, with the threshold being the putative mechanism by which SC activity could conceivably undergo a qualitative change in its capacity to elicit movement.Overall, the analyses do not seem to fully align with the conclusions drawn from them. The number of neurons included is small, with considerable variation in firing rates, and further, more substantial analyses on how activity relates to pre–saccade activity may require additional recordings. A related concern is about the key analyses that seek to establish the 'motor potential' of pre–saccadic SC activity through a correlation between activity and eye velocity. The authors acknowledge the challenge in this. The results rest on being able to determine when the saccade begins, and the reviewers raise concerns that this was indeed possible with the precision required in the subsequent correlations with neural activity. Reviewer 3 points out that eye velocities could establish which eye movements were pre–saccadic, but e.g. a lack of reporting of saccade size makes it difficult to judge whether this might work. The current set of analyses does not seem to convincingly demonstrate that pre–saccadic SC activity does in fact influence eye kinematics during BREMs.

We thank the editors and reviewers for the detailed critiques and comments on the previous version of the manuscript. It helped us realize that there were several problems with the paper, including misplaced priorities in the motivation and study description, lack of methodological reporting rigour, and clarity of the analyses. We have since completely reworked the manuscript, including the motivation and storyline, with primary emphasis on the motor potential of preparatory activity, which we think is strongly supported by the new, more detailed analyses. We hope the parts all reviewers found exciting in the previous version are as exciting, if not more. We have also included the threshold and rate analyses as is for the sake of completeness, but it is now at the end of the Results section.

We also attach some general comments that came from the consultation, that you may find helpful for revising your paper, and that clarify some of the reviewers’ original comments:A conceptual figure explaining the hypotheses, including a basic model of how SC, OPNs and burst generators interact, might be helpful for the reader.

Great suggestion, we now have a schematic circuit (Figure 1C) that explains the circuitry and specific hypothesis being tested.

Reviewer 1 asks about neuron properties in SC, but acknowledges not being an expert and this has therefore not directly impacted our assessment. The issue was indeed clarified by reviewer 3 who has experience in neural recordings. Reviewer 3 notes that variability such as reported here can be seen in the SC, and can strongly depend on different sized saccades and different directions of saccades. There are also classes of SC cells that show a strong motor burst only during the movement (with a very small temporal offset from movement onset), and maybe these will have less variability and show a fixed threshold. However, the analyses and details provided do not allow for assessing in more detail how such factors may have contributed to the results, and the small number of neurons seems to make it unlikely this can be addressed without additional data.

Thank you for informing us that the issue of firing rate range has now been clarified. We agree with reviewer 3’s assessment that the variability can be a function of many factors, including cell types, saccade amplitude and direction, etc. Briefly, we wish to emphasize the following:

The neurons reported in this study were all recorded from intermediate layers of SC, from sites that produced low latency microstimulation-evoked saccades. Most of our neurons were visuomovement neurons (47/50), and we have shown in a recent publication (Jagadisan and Gandhi, 2016) that visuomovement and movement neurons form a flexible continuum (pure “motor” neurons can sometimes show a visual response). We have now included a note (Materials and methods) about neuron types.

Cortical studies typically are based on a large number of neurons, partly because of the vast variety of discharge properties. Subcortical structures, in contrast, do not display the same level of heterogeneity. Thus, the cell count from these regions is generally smaller, especially studies that rely on the serial recordings with single electrodes. It is not at all uncommon for SC studies to be based on less than 50 neurons. For example, the Goossens and van Opstal studies on blink induced perturbations were based 25 neurons, and a recent study on saliency coding in SC was based on 34 superficial layer and 26 intermediate layer neurons (White, et al., 2017).

Regarding target locations: the target was placed at one of two locations, either close to the estimated center of the cell’s receptive field the center or at the diametrically opposite location in the other hemifield. We did not test multiple locations within the receptive field. We have now included a note on this (Materials and methods).

Regarding cell location within the SC: We examined whether the motor potential effect varies with rostralcaudal location of SC neurons. We found no effect (Pearson’s correlation between target eccentricity, a proxy for rostro-caudal location and peak motor potential correlation, control saccades: r = 0.31, p=0.06, blink-triggered saccades: r = -0.25, p=0.12, data not shown). Note that for each neuron, the target was presented in its optimal response field, and as such, there is no reason to expect variation with rostrocaudal position (unlike the case where the same saccade is performed for different neurons recorded along the rostro-caudal axis).

There are important links with other work, especially Kaufmann and colleagues and the work of van Opstal. Kaufmann, (2014) argue that 'motor potential' is a property intrinsic to motor cortical activity – opposite of the conclusion here about the SC. A key distinction between saccades and reaches might be that the trigger for reaches is 'upstream', perturbing the motor plan into potency, whereas for saccades, the trigger is implemented downstream, via the OPNs. Van Opstal and colleagues have suggested that each spike in the SC can contribute to saccade kinematics, but only consider spikes during movement. An additional "gating" mechanism is thus invoked to explain that many spikes occur outside saccades and without eye movements. The observation in your Figure 7 that spikes outside a saccade may contribute to movement properties seems relevant in this regard, but the analyses in this figure aren't supporting this as clearly as suggested.

Thanks for bringing up the connections with other recent studies. We completely agree with the assessment that these results, on first glance, seem to contradict the seminal observation by Kaufman, et al., 2014, and subsequently Elsayed, et al., 2016 that the ability to drive movements is an intrinsic property of motor cortical activity. It’s a key reason why we believe this study is of broad interest and therefore should be published in a high impact journal. As suggested in your feedback, this discrepancy may be because of some fundamental distinction between the skeletomotor and oculomotor circuitries, such as the location of the trigger in the pathway. However, we still do not want to rule out the possibility that the presence of a motor potential in premovement activity may be a property of sensorimotor systems in general.

While it is possible that the activity may need to be in a certain population subspace in order to trigger the movement (whether by opening a gate or efficient relaying of activity downstream, etc.), it remains possible that once the gates are opened via other independent means (e.g., the blink perturbation in this study), upstream activity canbecommunicated to the muscles to generate the movement, regardless of the properties of the activity (i.e., subspace occupied by the activity). It is difficult to know whether this is in fact a generalizably true feature without causal experiments in which the trigger is prematurely released. Consider also that in its latest avatar, the threshold hypothesis can be reframed in a manner similar to the optimal/potent space ideas (see attached Author response image 1)

**Author response image 1. respfig1:** Movement initiation models in the population dynamics framework. In all cases, population activity is represented as an evolving trajectory in multi-dimensional space – only the activity of 3 neurons/3 latent dimensions are displayed. (**a**) The optimal/potent subspace hypothesis suggests that movements are triggered when population activity reaches a pre-determined activity subspace (ellipsoid). Thus, only the blue trajectory will result in movement generation whereas trajectories that don’t reach this “optimal” subspace (red trajectories) don’t lead to movements. (**b**) The classic version of the threshold hypothesis posits that movements are initiated when the activity of individual neurons reaches a threshold level (3 different levels shown in left column). When represented collectively in a population framework, this is equivalent to a point in multidimensional space, which must be crossed to generate a movement, making it a far narrower “optimal subspace” equivalent. (**c**) Recent iterations of the threshold hypothesis expand the classic version to allow for a threshold at the population level, represented by a linear combination of individual neuron activities. This is equivalent to crossing a hyperplane in multi-dimensional space. Note that in all these cases, activity must cross some decision boundary (subspace bounds, point, or plane) in order to generate a movement (blue trajectories), implying that the set of activities prior to this point lacks the potential to do so.

Thus, its breakdown (and the associated emergence of a motor potential before the initiation criterion is reached when the trigger is released) supports the possibility that similar mechanisms may be at play in other motor systems as well.

The work of Goossens and van Opstal is also relevant in this regard. The mini-vector model proposes that each spike in SC contributes to a fixed length displacement of the eye during the window when gating is open and has been tested with blink perturbations. But it is unclear how this translates to the motor potential we observe, either before or during the saccade, since we are looking at correlated variability in SC activity and eye kinematics across trials. In fact, in Figure 5, this correlation is computed between activity and residual velocity projected onto the direction of the saccade target after subtraction of the blink-related eye movement template. This can cause the kinematic variable to be instantaneously negative (i.e., going away from the saccade target) on some trials, but as long as it is less negative on trials when the activity is higher, we still say that the activity has motor potential. The mini-vector model will predict that the eye moves in fixed vector increments towards the saccade target (which also roughly happens to be the optimal vector of the recorded neuron). Moreover, in Goossens and van Opstal, 2006, not all neurons show the fixed spike count property, and at the population level, spike counts on perturbation trials are slightly higher than on control trials. This is entirely consistent with our observation that spikes can leak through when the gating is open (leading sometimes to excess spikes as in Goossens and van Opstal, 2006).

We discuss both these points at length in the Discussion section in an attempt to reconcile these parallel views of movement initiation and execution.Reviewer #1:Summary:This study tests the influential threshold account for saccadic movements. The main finding is that SC neurons do not need to reach a fixed activity threshold to trigger a movement. The authors derive this from showing a latent motor potential in the preparatory activity of SC neurons, which they uncover by removing the inhibitory influence on SC neurons through blink–triggered saccades.The study is interesting, and speaks to a long–standing debate about the neural mechanism underpinning movement planning and initiation. The idea that activity has to reach a fixed threshold has gained a lot of attraction in the field, and this study questions that such a fixed threshold is indeed required. The use of blink–triggered saccades as a perturbation as used here seems elegant.My enthusiasm is somewhat dampened by the complexity of the paper and by being unsure about some of the effects and the interpretation.

Thank you for these kind comments. The manuscript has since undergone a complete rewrite and we have now made significant efforts to both simplify and clarify the results.

Firstly, if I'm correct, the authors seek causal evidence for movement occurring below threshold, by disinhibiting saccade activation via BREMs? Is this specific to the preparatory process?

Yes, we believe the reviewer’s insight is correct. We sought to evoke blinks during the reaction time period, after the ‘go’ cue but before the typical reaction time for a saccade. We consistently observed that the BREM is accompanied by a saccade to the target, and the movement occurs at a reduced latency than normal saccades (red dots in Figure 3A). Furthermore, such movements are as accurate as control saccades (Figure 3B).

In our view, the reaction time period is the preparatory period, and the underlying activity is considered to reflect a preparatory process. Our two central observations – disinhibition of the saccadic system with a blink perturbation during the preparatory period produces an eye movement, and the low frequency neural activity during the preparatory period is correlated with eye velocity – indicate that this preparatory activity is capable of evoking a movement and therefore has motor potential. These results contradict the notion that the movement preparation and movement execution signals are distinct and encoded serially.

At one point the authors set out to test "how the brain prepares for movements before issuing the command to initiate them". This seems different from asking whether preparation and execution might occur in parallel or not, which the abstract's final sentence alludes to. This seems relevant to judge whether the data support the conclusions the authors make, which alternate between the threshold theory, and preparatory activity influences.

Thanks for pointing out the disconnection between the two questions we posed in the last version of the manuscript. We have attempted to integrate them better in the revised version. We are indeed after causal evidence for whether movements can occur beforea standard initiation criterion is reached, and more importantly in this version, whether low frequency preparatory activity can leak through to affect movements (which is related to the question of whether preparation and execution are represented concurrently in neural

activity). We have also replaced the above quoted phrase with “how planning activity is appropriately parsed in order to prepare and execute movements” Introduction.

Second, there does not seem a behavioural manifestation of preparation. Was there any?

We apologize for the lack of clarity, but the question of motor potential during the reaction time period is in essence the question of whether there is any behavioural manifestation of movement preparation. This point is now emphasized in the manuscript (Introduction, Results sections). In terms of findings, the occurrence of premature, low latency blink-triggered saccades is one behavioural manifestation of preparation. The main result, i.e., the correlation between neural activity and eye kinematics before saccade onset is also an example of behavioural manifestation of preparation.

Third, are the two possibilities outlined in the Introduction indeed mutually exclusive? Could a parallel system not be equipped with a threshold mechanism, whereby there is continuous and parallel influence of preparatory signals, but even then, there is an overall activity level that will unleash the movement? The threshold theory states that a movement is triggered when a specific threshold is reached – at least in the standard cases usually tested. But it does not say that activity prior does not possess motor potential. A sprinter would normally start running when a go signal occurs, but he could start at any time prior to that. There is simply a good reason to inhibit that tendency because he would get disqualified otherwise. Does a threshold perhaps arbitrate between speed and accuracy?

This is a good point and one that’s at the core of our study. True, the threshold theory does not explicitly state that the activity prior to reaching threshold does not possess motor potential. But that is the most common interpretation of the threshold theory, that downstream motor circuits are not activated (and possibly not activate-able) before premotor activity reaches a criterion level. This interpretation has been implicitly assumed in several papers that refer to cognitive processes such as target selection and others associated with low frequency activity, and motor execution, as serial stages leading to behaviour (several publications from the Schall group).

It is also possible that under normal circumstances, a threshold may help balance speed and accuracy, as you suggest, but we find that eye movements are just as accurate when prematurely triggered by the reflex blink before reaching threshold. We now discuss this point in the manuscript (Discussion section).

Fourth, I'd like to be assured that the results indeed support the conclusions. There is some under–reporting of procedures, but it is also hard to gauge some of the neural data given the different transformation conducted. I list some of these under minor comments.One point I'd like to clarify, noting that I am not an expert on SC firing rate properties. The key results are reported over a very short time window of e.g. 20ms. How then can spike rates per second of 400 or more be inferred robustly? The authors use a convolution approach but it would be reassuring to see raw firing rates here, and to be convinced that the results do not contain any spurious values from the convolution. But even if this is not the case, out of the neurons reported some have spikes/s of 10–20, others of 400+. Is this within the normal range of firing rates in SC neurons coding for saccades? And even if, I wonder how with this range all neurons can indeed relate to velocity so closely?

We thank the reviewer for identifying his/her knowledge-base for SC physiology. Please allow us to reiterate the key points we made to the Senior Editor above:

SC firing rates indeed span a large range but there is a lot more homogeneity in the firing rates of subcortical neurons than of cortical neurons. This allows scientists to make conclusive interpretations based on fewer cell counts.

In terms of firing rate analysis, the conventional approach is to convert the discrete spike train for each trial into a continuous “spike density” waveform through convolution with either a Gaussian (3ms standard deviation used in the current study) or a function resembling an EPSP. All operations (e.g., averaging, normalization, alignment, binning into small windows) are performed on such spike density waveforms. When using this method, it is not at all uncommon to find neurons that exhibit a low-frequency prelude on the order of 10-20 spikes/s (or as high as 100-200 spikes/s), which then turn into a high-frequency burst with peak firing rate well over 500 spikes/s. Previous studies have compared firing rate parameters from such spike density waveforms with PSTHs based on interspike intervals, and have found comparable results.

Reviewer #2:In this paper, Jadagisan and Gandhi argue against the old but popular notion that movement initiation occurs as a consequence of preparatory activity exceeding a fixed threshold. Focusing on saccade preparation in the superior colliculus, they evoke saccades with lower–than–normal reaction times by eliciting the trigeminal blink reflex – a response known to lead to inhibition of omnipause neurons (OPNs) that inhibit eye movement. SC activity prior to these blink–evoked saccades is clearly lower than prior to normal saccades, contrary to a putative fixed threshold. Furthermore, activity prior to saccade onset appears to influence the kinematics of eye movements during the blink, suggesting that the preparatory activity has the potential to directly influence movement if allowed to via inhibition of the OPNs.I am sympathetic to many of the ideas in this paper. However, I have reservations about the strength of the results and conclusions. First, refuting a fixed threshold for SC activity to elicit a saccade does not seem like a surprising result, particularly in light of papers cited by the authors which also claim that the threshold may be flexible or task–dependent. The more novel claim is therefore that pre–saccadic activity has 'motor potential' that can be unmasked by the inhibition of OPN activity during the trigeminal blink reflex. However, I have some significant concerns as to the veracity of this result.

Thank you for these comments and the support for the paper’s philosophy. Many of your critical observations have helped us realize the logical holes in explaining some of the results in the previous version. The manuscript is now completely rewritten, with new analyses and figures, and we have hopefully made every attempt to address the concerns about the veracity of the result.

'No threshold' versus 'Flexible threshold'A central claim of the paper is refuting the notion that SC activity must reach a specific threshold in order to trigger a saccade. However, the manuscript is inconsistent on whether it purports to demonstrate the absence of a threshold, or simply a flexible one. The rhetoric seems to frequently switch between these two positions. For instance, the abstract claims that "the results bring into question threshold models for saccade generation" while the significance statement states that "the threshold is not fixed, but can be flexibly modulated based on the level of inhibition". This incoherence persists throughout the manuscript (e.g. subsection "Blink–triggered saccades are evoked at lower thresholds compared to normal saccades" vs "it is NOT necessary for activity in SC to reach a threshold level in order to produce a movement"). I don't believe the data can disambiguate between a flexible threshold and no threshold.

Thank you for pointing out this fundamental problem when talking about thresholds. The popular notion of a threshold implicitly assumes that the threshold is fixed and rigid, which is what we wanted to test in this study. Saying that the threshold may be flexibly modulated is just another way of saying that there is no fixed threshold, since one could still argue that any activity level attained before a movement is initiated is a “threshold”, whether it is remains consistent or not. We do not claim to disambiguate between flexible and no threshold, only between fixed (necessary condition) and flexible (not fixed, not necessary) threshold. We hope the rewritten manuscript is now sufficiently clear in dealing with this issue, although it has taken a backseat with the increased emphasis on motor potential.

Evidence of 'motor potential'Given that the notion of a flexible threshold is not altogether novel (albeit it is a nice, clear demonstration), the claim that 'sub–threshold' SC activity can influence movement becomes critical to the paper.

We agree, and hope this version reflects the prioritization of the more novel “sub-threshold motor potential” result.

The authors establish the 'motor potential' of pre–saccadic SC activity through a correlation between activity and eye velocity. A major problem with this approach, however, as the authors acknowledge, is that in order to determine which activity is pre–saccadic, they need to identify when the saccade begins, but this is difficult to determine precisely when there is also a blink–related eye movement. It does not seem unlikely that in certain trials, the 'true' saccade onset time might have been earlier than estimated, in which case it would not be surprising for the "pre–saccade" activity to correlate with "pre–saccade" eye velocity (since both are actually post–saccade–onset).

We agree with these concerns based on the presentation of the analyses in the previous version. To be honest, we were also initially concerned whether we were being misled by the blink perturbation and detection of the saccade embedded in the combined blink-triggered movement. Ideally, one would want to use a direct way of inhibiting the OPNs to avoid these potential pitfalls, but considering that the blink perturbation has previously demonstrated its utility as a powerful, non-invasive tool that offers a behavioural readout of underlying processes (Gandhi and Bonadonna, 2005; Walton and Gandhi, 2006; Katnani and Gandhi, 2013; Jagadisan and Gandhi, 2016) we decided to stick with it for this study. In this version, we performed more finegrained analyses of instantaneous correlations along with control analyses, which we think should alleviate some of these concerns (see responses to the next couple of comments).

Note that the motor potential result on control trials (and thus after saccade onset on blink trials) is by itself a novel finding, since no previous study has directly demonstrated the correlated variability between SC activity and saccade kinematics across trials, even though there’s a general sense in the field that the stronger the burst, the faster the movement. The studies that come close to explicitly to showing this, in our view, are the ones with the mini-vector displacement model. But it is still a leap to go from there to instantaneous correlation between firing rate and kinematics. This point was not clear in the previous version, but now is presented in the manuscript (Discussion section).

Nevertheless, we think that it is unlikely for the detected time of saccade onset to be too far from its true value. The velocity profiles shown in Figure 2A are largely representative of what happens on individual trials – in these examples, it is evident that since we use the minimum of the horizontal and vertical movement deviation times from the BREM template, if anything, we are estimating the onset to be *earlier* than what one might expect if computing onset using the combined vectorial velocity profile.

A related issue is that saccade onset is determined based on when eye velocity exceeds a threshold z–score (based on the distribution of blink–related eye movements). Given the smoothness of the eye movements, the eye velocity in the 20ms preceding the detected onset of the saccade is bound to be correlated with the peak eye velocity. (e.g. see Figure 7A, lower panel, top line; because the saccade has negative velocity, it is inevitable that eye velocity will be negative as it breaks the initiation–detection threshold). So, then the correlation between 'pre–saccade' activity (which will be correlated with activity just after the saccade starts) and 'pre–saccade' eye velocity becomes inevitable.

We would like to suggest that this is actually not a concern, since are now using the residual (relative to the BREM) velocities in the blink condition. Because the saccade is largely directed towards the target after saccade onset, variability across trials is far higher during the saccade than before its onset, and not necessarily correlated with each other. This is seen in the traces in Figure 5A, where there is a discontinuity at time 0 (detected saccade onset), where all the traces come together. We tested for a systematic relationship within the velocity traces before and after saccade onset by performing a correlation analysis similar to the motor potential analyses, except on the velocity traces with themselves, and we found none (see attached Author response image 2).

**Author response image 2. respfig2:** Self-correlation of velocity profiles. Left – Across-trial correlated variability of instantaneous velocity on control trials. These correlations are computed similarly to the activity-velocity correlations in Figures 4C and 5C, except with the activity replaced by velocity. Smooth and continuous velocity profiles result in a window of high correlations (gray square) where the velocity values are correlated amongst themselves. Right – For blink trials, the velocity profiles are correlated with each other over time after saccade onset (gray square), and separately, before saccade onset (black square, during the epoch of preparatory motor potential), with a discontinuity in between (narrowing of correlations along the diagonal). It is important to note that the correlation between pre-saccade and post-saccade velocities is not nearly as strong (if they were, one would expect a window of high correlations marked by the dashed red square). Moreover, if saccade onset were over- or under- estimated, either the black or gray windows should’ve crossed the zero point.

Note the narrowing of the correlation boundaries around time 0, indicating that the velocity traces before and after detected saccade onset have different correlation structures. A related phenomenon is seen in Figure 5C, where there is a narrowing discontinuity in the motor potential correlation at time 0 on the x-axis (detected saccade onset). If saccades were simply systematically estimated to be starting later than their true value, then the correlation profile in this fine-grained Figure would look smooth crossing the t=0 axis, which is not the case. This property can also be seen in Figure 5E, where there is an abrupt discontinuity/dip at time 0.

To guard against these problems, the authors analyze a window of activity even further from saccade onset ([–40,–20]ms, as opposed to [–20,0]ms). The problems with analyzing this earlier window are that (i) the correlations between activity and velocity are extremely weak, (ii) for the example neuron, the eye velocity associated with SC activity seems to be negative in most trials, hardly demonstrating 'motor potential', and (iii) it is by no means clear that moving the window forward by 20ms alleviates the confounds described above (i.e. the faint correlations may still be driven by accidentally catching peri–saccade activity in enough trials, or because of smoothness of the eye velocity signal as it crosses threshold). I believe the authors need to provide a much stronger argument than that that their primary result is not undermined by these potential confounds.

It is true that the correlations before saccade onset are weaker than after, but that is to be expected since the system is not fully engaged in driving the movement until it goes into the high frequency burst mode. But the fact that we do see these correlations consistently, starting 25-30ms before saccade onset is something we find remarkable.

Another strong piece of evidence for the fact that saccade onset is not corrupting our analysis is the fact that the correlations before saccade onset are only seen, at the optimal efferent delay of 12ms, when we use the residual velocity projected in the direction of the saccade target as our kinematic variable (Figure 5), and not when the projection step in omitted (supplementary figure 2). The rationale for taking the projection for our main analysis was that variations in BREM kinematics in directions orthogonal to the direction the neuron cares for (saccade target vector) may mask or eliminate any underlying correlations. If saccade detection were an issue, and the saccade had already started during what we claim is the pre-saccade period, the correlation should be present before time 0, at the optimal delay, even in supplementary figure 2 (since it is actually postsaccade, which we know has motor potential from the other analyses). But it doesn’t.

Thus, we have strong reasons to believe that the observed results are not the spurious side effects of our detection algorithm. We thank the reviewer for pushing up on this point, and we hope (s)he is favourably swayed by our updated analyses. We have clarified these points and arguments in the document (Results section).

Reviewer #3:The authors used a blink perturbation to shift visually guided saccades towards having lower latencies than usual. This allowed measuring SC activity for these "premature" saccades. SC neurons burst for the premature saccades, but the saccades were elicited at lower firing rates. Also, the low–level activity in the SC "leaked" into downstream structures with the perturbation, and affected eye movement trajectories. The authors discussed these results in relation to rise–to–threshold models of saccade triggering, arguing that a fixed threshold is not necessary for movement generation. The authors interpreted the "leakage" as the pre–saccadic building possessing a motor potential.The paper is generally quite well written (particularly the introduction), but there are notable important deficiencies in clarity in several places highlighted in the more detailed comments below. I would strongly suggest that the authors provide more details in general throughout the manuscript (e.g. neuron depths, locations in the SC topography map, and saccade sizes and directions etc).

Thank you for requesting this information. Briefly:

All neurons in our database are visuomovement neurons, thus they reside in the intermediate/deep layers. This is now made clear in the Materials and methods.

Targets were placed either in/near the estimated center of the cell’s movement field (thus matching the cell’s optimal saccade vector) or at the diametrically opposite location. The analysis reported here is only for the subset of trials for which the target (and saccade) were in the response field. This is also now made clear in the Materials and methods.

The optimal vectors of the neurons ranged from 9-25 deg in amplitude and spanned all directions. Thus, we covered a large part of the central portions of SC (Materials and methods).

In terms of the study itself, I found that the results of Figures 6 and 7 are the most important and most interesting aspects of this study. I would highly recommend that the authors expand on these results even more, if possible (even at the expense of the earlier results, which are a bit less exciting).

Thank you for the suggestion. We have revised the manuscript drastically according to this and other reviewers’ suggestions, with a much stronger emphasis on the motor potential result.

For example, do these correlations get modulated with rostro–caudal position of the neurons?

Thank you for this question. We examined whether the motor potential effect varies with rostral-caudal location of SC neurons. We found no effect (Pearson’s correlation between target eccentricity, a proxy for rostrocaudal location and peak motor potential correlation, control saccades: r = 0.31, p=0.06, blink-triggered saccades: r = -0.25, p=0.12, data not shown). Note that for each neuron, the target was presented in its optimal response field, and as such, there is no reason to expect variation with rostro-caudal position (unlike the case where the same saccade is performed for different neurons recorded along the rostro-caudal axis).

How do these observations relate to the "mini–vector" models of Van Opstal? What prevents "leakage" without the perturbation? etc. The description felt a bit light even though I find this result much more interesting than the threshold stuff.

We now discuss this point in detail. See section entitled “The role of SC and brainstem in saccade execution” in Discussion section.

Another question is how the threshold results relate to pure saccade bursters. What would their burst size look like for the short latency saccades after the blink manipulation? And, if it is not affected, then what does this imply for their interpretation?

Good question, but we are unsure whether the reviewer is referring to pure saccade bursters in the SC (“motor neurons”), or the burst neurons in the reticular formation.

If referring to the SC motor neurons, since the majority of the neurons in this study are visuomovement (47/50), we do not have enough “pure” bursters to perform this analysis. However, we have previously shown that visuomovement neurons and motor neurons form a continuum (Jagadisan and Gandhi, 2016). Therefore, we believe their burst size, just like those of visuomovement neurons (Figure 6A), will be unaffected.

If referring to the brainstem, previous work has shown that the discharge profiles of burst generator neurons in the reticular formation are a scaled version of the observed eye velocity waveforms (Cullen and Guitton, 1997). Moreover, in the presence of a perturbation – for example, when a natural blink accompanies a saccade (Gandhi and Katnani, 2010) or when a brief torque is applied to the head during eye-head gaze shifts (Sylvestre and Cullen, 2006) – burst neuron activity is also modified to account for the observed changes in eye velocity. Given these results, we predict the lower brainstem burst generator neurons will exhibit low frequency activity to produce the slow eye movement leaked by premature inhibition of OPNs followed by a high frequency burst that generates the saccade. We have included this note in the manuscript (Discussion section)

– Introduction: I think I understand what you mean, but the way things are phrased, it might appear to contradict the abstract. In other words, in the abstract, you say that the preparatory activity has motor potential. Here, in the paragraph of the Introduction, you describe the two possibilities: (1) parallel, (2) serial. In (1), the preparatory activity would have the "motor potential" as you say in the abstract. In (2), the potential is acquired only later. So, my problem comes when you say "…presence of such a motor potential…". Given the positioning of the definition of motor potential in the previous paragraph, the use of "such" here makes the reader think that you will show in your results scenario (2) above instead of scenario (1), which would contradict the Abstract. I would rewrite the sentence “How might we test for the presence of such a “motor potential” in low frequency neural activity? “to be more explicit and avoid confusion of a possible contradiction with the Abstract (expand the sentence and remove "such").

Good point. We have now rephrased this part of the Introduction such that the parallel/concurrent view is discussed last in the previous paragraph, making the text flow more naturally into the next paragraph (Introduction)

– Introduction and Figure 7: I think these results are the most interesting aspect of this study.

Thank you. We agree, and all the reviewers’ comments and suggestions to this effect has led to a drastic redesign of the manuscript.

– Results, I think for a broad journal like this one, "the delayed saccade task" is way too brief a description. Please explain the task briefly here (even if you do it later in the methods section, which you actually did not do sufficiently either). Statements like "after the go cue" would be foreign to a great majority of readers.

This version now includes a more detailed description of the task (Results, Materials and methods).

– Figure 2 legend, please define BREM even if it was defined in the text. Some readers quickly parse through figures before reading the text, and they would not know what BREM means.

Thanks for pointing this out, we have now expanded BREM where necessary.

– Figure 2 legend: do you mean "relative to movement onset" or "relative to blink onset"? Similarly, for "after movement onset". I assume you mean "after blink onset". Please clarify. This is very confusing especially because there are different types of "movement" in your paradigm.

For blink-triggered movements, blink onset leads to the BREM, which is the onset of the overall movement (as opposed to just the saccade). But the point is taken since we often use “movement” interchangeably with “saccade” throughout the manuscript, and we have now replaced “movement onset” in the legend with Figure 2B “overall movement onset”.

– Figure 2: the colours in panel B are not defined in the figure legend (and very late in the main text – also, in the text, the justification for the division into the two colours is not clear at all).

We have now removed the different colours and have included a line to mark the division between trials used for different analyses (Figure 2B).

– Results: what's the point of Figure 2B? I think that readers should be walked through the logic of your manipulation and why you feel the need to show Figure 2B, and why there are two different colours etc. Also, the key is really Figure 3, and both Figures 2 and 3 are more introductory and confirmatory than anything else. I would suggest combining Figures 2 and 3 (as one related methods figure) and explaining clearly in the text how your manipulation leads to your goal of getting short RT's. This is an example of a strong deficiency in clarity of the text.

Agreed, and these points are now clarified in the text (Results section). But since the content of Figures 2 and 3 are qualitatively different (saccade detection method vs low latency and accuracy results), we have opted to retain them as separate figures.

– Subsection “The blink perturbation triggers reduced latency saccades”: what does "with saccade onset greater than 20ms" mean? This part of the text is very unclear.

We have expanded and clarified this now Results section.

– Subsection “The blink perturbation triggers reduced latency saccades”: normalized to what?

The activity was normalized to the peak trial-averaged firing rate during normal saccades. This is now stated explicitly (Results and Materials and methods sections).

– Figure 4B: please indicate in the legend that the n=50 is the number of neurons, but that you show more points in the figure. My understanding is that you show 50 points for each time window highlighted in A, correct?

The reviewer’s understanding of the points in this figure is correct. We have now clarified this in the figure caption Figure 6B.

– Figure 6, please indicate the sizes and directions of the saccades, and especially in panel A.

Panel A in the (new) Figures.4 and 5 contains velocity traces from one session and is thus just an example. We have now included a note on the saccade sizes and directions used in this study in the Materials and methods section.

– Subsection “SC preparatory activity preceding the saccade-related burst possesses motor potential”: pointing to methods "for more details" seems insufficient here. Explain to the readers more especially because this is critical for the most important aspect of the study.

We have now completely changed the way we analyze the motor potential data and have incorporated more details in the text (Results section).

– Subsection “SC preparatory activity preceding the saccade-related burst possesses motor potential”: again, similar idea. why 15ms? etc

The 15ms time point was chosen arbitrarily, as an example, to show the scatter for 3 time points before, at, and after the onset of the saccade. The 12ms efferent delay was chosen due to the estimated optimal delay between activity and velocity from subsequent analyses. The actual analyses are performed for all movement time points and a range of efferent delays, and the chosen example time points in panel b in highlighted by an asterisk in the main figure (Figures 4C and 5C, Results section).

– Subsection “SC preparatory activity preceding the saccade-related burst possesses motor potential” needs more clarification

We have expanded the text to clarify this point (Results section, Materials and methods section).

– Description of Figures 6–7: I didn't see the p–values associated with all these correlations. Please add them.

We have now included a better description of the statistical analyses and included p-values for these results (see Results and Materials and methods).

– Subsection “Implications for threshold-based accumulator models”: is it established that OPN's implement strict gating?

Yes, OPN activity is known to be tightly linked to saccade initiation (Cohen and Henn, 1972; Keller, 1974).

– Materials and methods and paper in general: there are a lot of missing details. Where did you record in the SC both in terms of rostro–caudal/medio–lateral position and also in terms of depth? How large were the saccades? What were their directions? Is there dependence of the effects on all of these SC and saccade parameters?

We are including a comment made to a point raised by the editor above:

Regarding target locations: the target was placed at one of two locations, either close to the estimated center of the cell’s receptive field the center or at the diametrically opposite location in the other hemifield. We did not test multiple locations within the receptive field. We have now included a note on this Materials and methods section.

Regarding cell location within the SC: We examined whether the motor potential effect varies with rostral-caudal location of SC neurons. We found no effect (Pearson’s correlation between target eccentricity, a proxy for rostro-caudal location and peak motor potential correlation, control saccades: r = 0.31, p=0.06, blink-triggered saccades: r = -0.25, p=0.12, data not shown). Note that for each neuron, the target was presented in its optimal response field, and as such, there is no reason to expect variation with rostro-caudal position (unlike the case where the same saccade is performed for different neurons recorded along the rostro-caudal axis).

[Editors’ note: the author responses to the first round of peer review follow.]

Reviewer #1:This is an interesting, careful study looking at responses in the superior colliculus (SC) and their contribution to saccade initiation and to the metrics of the evoked eye movements. The use of blink–triggered saccades provides a valuable tool for manipulating the dynamics of the oculomotor circuitry in a relatively natural yet informative way. The results provide important evidence showing that the subthreshold, preparatory activity typically seen before saccades partially specifies the motor command itself, as reflected in saccade kinematics. Furthermore, they demonstrate that the threshold for triggering a saccade is, at the very least, more flexible than is typically assumed. The findings are significant because the concept of a fixed threshold has become dogma in studies of decision making, yet relatively little is known about it in neurophysiological terms, even for the clearest instance – precisely that of saccade generation.

Thank you for these nice observations.

My only criticism is that the way the conclusion about the threshold is articulated is a bit too narrow; perhaps misleading. To begin with, it is unclear what or where exactly the threshold is, in neural terms, so the assertion that "reaching threshold is not a necessary condition for movement initiation" (Abstract, Introduction, subsection “Blink-triggered saccades are evoked at lower thresholds compared to normal saccades”) seems extreme, and at some level contradictory. For instance, the blink might simply decrease the required threshold level. Also, the term "threshold" in a way describes the ballistic nature of the saccade; one could argue that, empirically, the threshold is simply whatever happens that turns the current motor activity into an uncancellable motor command – but it is unclear what exactly this is, or whether the threshold is really a collective property to which, say, LIP, FEF, and SC all contribute. What is clear is that the threshold is not fixed, and that threshold crossing may have something to do with inhibition of the OPNs, as the authors already imply. The Discussion shows that the authors appreciate the difference between existence and flexibility, but I think rephrasing this main conclusion in a more nuanced way at the various points at which it is mentioned would improve the clarity of the main message.

We thank the reviewer for catching this subtle point, which may be confusing to some readers. It is true that there is some ambiguity in what we mean by “threshold”, especially in the context of “reaching threshold is not a necessary condition for movement initiation”, given that the term itself is not well-defined. One way to define it, as the reviewer points out, is to say that the threshold is simply whatever level of activity exists at the time ongoing activity turns into an uncancellable motor command, but this study is ill-suited to know where or whether such a point-of-no-return exists. Thus, the only thing one can do is to test whether reaching a “fixed” threshold is a necessary condition for movement initiation, and our approach was to consider any activity level before saccade onset in the control condition as a putative threshold level and compare the activity at corresponding times on blink-triggered saccade trials with that level. This way, we believe we convincingly demonstrate that a “fixed” threshold criterion cannot exist.

We are on board with the reviewer in thinking that it is entirely possible that the “threshold”, defined in a different way, can vary or be flexible across conditions or even trials, except that we fail to appreciate the utility of such a definition. Why give it a label if it is not going to be “one thing”?

This disconnection between whether a threshold exists and whether it is fixed is addressed at various points in the manuscript, as the reviewer acknowledges. In places where we state that “reaching threshold is not a necessary condition for movement initiation”, we use “threshold” as shorthand for its original, simplest interpretation – that of a fixed spike rate that must be crossed before movements can occur. However, given the current literature of dynamic and time-varying thresholds, we admit that there is ambiguity in using such a term, and we have now clarified what we really mean by specifying a “fixed threshold” where appropriate (Abstract, Introduction, subsection “Blink-triggered saccades are evoked at lower thresholds compared to normal saccades”).

Reviewer #2:Overall, the authors have done an outstanding job revising the manuscript. The manuscript now has a much clearer conceptual focus, and more rigorous and convincing analysis. I feel this is now a strong and compelling paper. I have no major outstanding concerns.

Thank you for these kind comments.

Reviewer #3:I like this revised version much. It's very nice. It is clear for the most part, and it has addressed the previous comments. I have only a few suggestions for improvements:– Results section: some parts are difficult to understand. I would explain the divide between the <20 and >20ms movements better. More importantly, later in the same set of lines when you refer to "motor potential analyses" and the "threshold analyses", these two concepts are a bit foreign still, and just saying "the threshold analysis" or "the motor potential analysis" assumes that readers know what "the" analyses are. I know that you alluded to them earlier in the introduction, but it still helps to rewrite this final sentence in a more expositive way especially so early in the paper. For example, perhaps you can say something along these lines: because blink–triggered movements with saccade onset >20ms after blink onset likely reflect the idea that they were triggered anew by the blink–related disinhibition, they would be ideal for exploring whether there is a motor potential in the period even well before the actual saccade onset. On the other hand, for movements with <20ms, these are movements for which motor preparation was well under way, and are thus ideal to explore whether triggering can happen even before a fixed threshold is reached. (In other words, explain a bit more verbally the different "analyses" that will come up later in the paper. My exact wording above is probably bad but you can come up with the perfect wording).

Thank you for this suggestion. We have now reworded and expanded these statements in the Introduction accordingly (Results section).

– The motivation for Figure 3 is still somewhat unclear to me. Doesn't it just replicate previous results? If so, then why show it as a main figure? Perhaps it could be included as a supplementary figure at an appropriate time later when you want to address a possible confound in interpreting motor potentials or what have you? It's not entirely clear why the reader needs to go through a new replication of older results from the same lab. I guess the point to make is in the Results but if this point can be made with citations, then this is just fine.

We did debate whether to include this figure in the manuscript, but decided to do so anyway for three reasons – (1) the main result in Figure 3A about the reduction in latency for blink-triggered saccades was shown by Gandhi and Bonadonna, (2005) more than a decade ago, and it is unclear that too many people, even in the oculomotor field appreciate the crux of that result. Moreover, the figure in the earlier study did not show a background distribution for control saccades, so we felt this visualization was useful. (2) We use a method of extracting the saccadic component from blink-triggered movements that has been used previously in our lab (Katnani and Gandhi, 2013), but not in the context of reduction of latency for single targets as in the 2005 paper. Thus, we felt it was necessary to demonstrate that the previously reported trend of saccade reaction time with respect to blink time is preserved even using this method. (3) Figure 3B is also known from previous studies (Goossens and van Opstal, Gandhi and Bonadonna), but one reviewer had asked about the potential confound of accuracy in the previous round of submissions, again highlighting the fact the blink-related results are not widely appreciated. Moreover, we specifically performed this analysis for the subset of trials where saccades occurred at least 20ms after the blink (the ones used for most analyses), something that is important to do to ensure that there is no difference between blink-perturbed (previous study) vs blink-triggered (this study) saccades with respect to accuracy.

– Subsection “SC preparatory activity preceding the saccade-related burst possesses motor potential”: better say "after time 0 in the x–axis in the heat map in Figure 5C" because both x and y axes have time in them, so it's better to clarify which "time 0" you mean.We have now clarified this.– Please explain Figure 5B in more detail in the text. It is not described at all in the text, but the reader would want to know the difference from Figure 4B. So, just walk us through a textual description of Figure 5B and what it means for your interpretation and when comparing to Figure 4B. What are we supposed to understand from looking at this Fig? This would also be a nice segway into describing Figure 5C. Currently, I'm not sure what the take home message from Figure 5B is based on reading just the results text.

Figures 4B and 5B are meant mainly to illustrate an example dataset with the scatter plot showing correlations, mainly to orient the readers to what the summary heatmaps in Figures 4C and 5C, respectively, show (to know what is plotted). They are not meant to be a primary result, since they are from just 1 neuron. Regardless, this is a good point, and we have made an effort to highlight important aspects (such as the presence of a correlation in the early and middle windows, esp., relative to Figure 4B) in the text (subsections “SC activity is correlated with saccade kinematics for normal saccades”,”SC preparatory activity preceding the saccade-related burst possesses motor potential”).

– In the section of Figure 5, you don't mention different rostra–caudal positions. I know that in the responses document, you say that there was a negative result, but this needs to be mentioned in the Results section and then maybe discussed briefly in the discussion. For example, you argue that the target was always the optimal target for the neuron. However, different neurons along the rostra–caudal extent of the SC will have different projection strengths to brainstem pre–motor neurons, right? So, does the speed of the pre–saccadic eye movement in blink triggered movements scale with the target eccentricity? This would lend strong support to your arguments in subsection “SC preparatory activity preceding the saccade-related burst possesses motor potential”.

Thank you for bringing up this important point. We performed this analysis and found that while the correlation between neural activity and velocity is not a function of target eccentricity (proxy for the neuron’s rostro-caudal location in SC given that the targets were presented at roughly the optimal location of the recorded neuron), the velocity of the movement was. To increase statistical power for this analysis and to account for variability in initial eye positions, we used the actual saccade amplitude on individual trials instead of the target location (which tended to be clustered at a few values), generating a “main sequence” for both the saccadic and pre-saccadic components. The pre-saccadic velocities also followed a monotonically increasing main sequence relationship, although it was not as strong as the saccadic relationship. This result is now shown in Figure 5 —figure supplement 2 and reported in subsection”SC preparatory activity preceding the saccade-related burst possesses motor potential..

– Subsection “Blink-triggered saccades are evoked at lower thresholds compared to normal saccades”: perhaps it could be nice to start with a summary sentence: “Our results so far show that… and…”. Then, you can say: “next, we used the fact… to study a related issue of what it takes to trigger the movement itself."Thanks for this suggestion – we have now included a linking sentence for smoother transition (subsection “Blink-triggered saccades are evoked at lower thresholds compared to normal saccades”).– Subsection “Blink-triggered saccades are evoked at lower thresholds compared to normal saccades”: here could be a place to address saccade accuracy (Figure 3) as a supplementary figure. For example, you can say that they are triggered earlier and with lower activity, even though they are still spatially accurate.

We have decided to include Figure 3 (please see our response to that point above), because we feel it is an important (and subtly different from previous studies) result that will aid the reader in visualizing what is going on.